# Cell–substrate adhesion drives Scar/WAVE activation and phosphorylation by a Ste20-family kinase, which controls pseudopod lifetime

Shashi Prakash Singh[1], Peter A. Thomason[1], Sergio Lilla[1], Matthias Schaks[2], Qing Tang[3], Bruce L. Goode[3], Laura M. Machesky[1], Klemens Rottner[2], Robert H. Insall[1,4]*

1 CRUK Beatson Institute, Glasgow, United Kingdom, 2 Zoological Institute, Technische Universität Braunschweig, Braunschweig, Germany & Cell Biology, Helmholtz Centre for Infection Research, Braunschweig, Germany, 3 Brandeis University, Waltham, Massachusetts, United States of America, 4 Institute of Cancer Sciences, University of Glasgow, Glasgow, United Kingdom

* Robert.Insall@glasgow.ac.uk

**Data Availability Statement:** All relevant data are within the paper and its Supporting Information files.

## Abstract

The Scar/WAVE complex is the principal catalyst of pseudopod and lamellipod formation. Here we show that Scar/WAVE's proline-rich domain is polyphosphorylated after the complex is activated. Blocking Scar/WAVE activation stops phosphorylation in both *Dictyostelium* and mammalian cells, implying that phosphorylation modulates pseudopods after they have been formed, rather than controlling whether they are initiated. Unexpectedly, phosphorylation is not promoted by chemotactic signaling but is greatly stimulated by cell:substrate adhesion and diminished when cells deadhere. Phosphorylation-deficient or phosphomimetic Scar/WAVE mutants are both normally functional and rescue the phenotype of knockout cells, demonstrating that phosphorylation is dispensable for activation and actin regulation. However, pseudopods and patches of phosphorylation-deficient Scar/WAVE last substantially longer in mutants, altering the dynamics and size of pseudopods and lamellipods and thus changing migration speed. Scar/WAVE phosphorylation does not require ERK2 in *Dictyostelium* or mammalian cells. However, the MAPKKK homologue SepA contributes substantially—*sep*A mutants have less steady-state phosphorylation, which does not increase in response to adhesion. The mutants also behave similarly to cells expressing phosphorylation-deficient Scar, with longer-lived pseudopods and patches of Scar recruitment. We conclude that pseudopod engagement with substratum is more important than extracellular signals at regulating Scar/WAVE's activity and that phosphorylation acts as a pseudopod timer by promoting Scar/WAVE turnover.

**Funding:** This work was supported by Cancer Research UK core grant number A17196 and Multidisciplinary Award A20017 to RHI, by the Deutsche Forschungsgemeinschaft, grant GRK2223/1 to KR, and a grant from the NIH (GM063691) to BG. The funders had no role in study design, data collection and analysis, decision to publish, or preparation of the manuscript.

**Competing interests:** The authors have declared that no competing interests exist.

**Abbreviations:** Arp2/3, actin-related proteins 2/3; cAMP, cyclic adenosine 3',5'- monophosphate; CYRI, CYFIP related Rac1 interactor; F-actin, filamentous actin; GDP, guanosine 5'-diphosphate; GTP, guanosine 5'-triphosphate; KO, knock out; LC-MS/MS, liquid chromatography-tandem mass spectrometry; MEF, mouse embryonic fibroblast; WASP, Wiskott-Aldrich Syndrome protein; WT, wild type.

## Introduction

Scar/WAVE is the dominant source of actin protrusions at the edge of migrating cells. In particular, lamellipods (in mammalian cells cultured in 2-D) and pseudopods (in cells in 3-D environments, or cells such as amoebas) are driven by Scar/WAVE recruiting the actin-related proteins (Arp2/3) complex, which in turn promotes an increase in the number of polymerizing actin filaments and growth of actin structures [1]. It works as part of a large, 5-membered complex, whose members have multiple names [2]; in this paper, they will be referred to as Nap1, PIR121, Scar, Abi, and Brk1 in *Dictyostelium*, and Nap1, PIR121, WAVE2, Abi2, and HSPC300 in mammals.

The principal known activator of Scar/WAVE complex activation is the small GTPase Rac. Inactive Rac is guanosine 5' diphosphate (GDP)-bound but on stimulation becomes temporarily guanosine 5' triphosphate (GTP) bound. The GTP-bound, but not the GDP-bound, form binds to the complex [3], in particular through the A site of PIR121, which includes a Rac-binding DUF1394 domain [4]. Interaction with GTP-bound Rac is essential for the complex to be able to function [3,5]. However, though it is clear that Rac is essential, it is not the only regulator [1]. Various experiments have found that Rac activation occurs later than the onset of actin-based protrusion [6], and signal-induced actin polymerization can occur earlier than Rac activation [7]. In *Dictyostelium*, Scar/WAVE behavior is much more locally variable than Rac activity [8], so the Rac cannot simply be driving the changes in Scar/WAVE. To understand pseudopod dynamics, it will therefore be vital to enumerate different modes of Scar/WAVE regulation.

One potential form of regulation—phosphorylation—has been described in a number of papers and is also found in untargeted and high-throughput screens. A typical narrative is that Scar/WAVE is phosphorylated in response to external signaling, through kinases such as (particularly) the global signal transducer ERK2. This has been described in cultured fibroblasts [9], mouse embryonic fibroblasts (MEFs) [10], and endothelial cells [11]. The phosphorylation is typically found to change the complex from an inactive to an activatable state, so Scar/WAVE phosphorylation directly leads to actin polymerization. Tyrosine kinases, in particular Abl, have been found to be similarly activating [12,13]. These reports are curious, for a number of reasons. First, actin is a strongly acidic protein, so phosphorylation of binding proteins typically weakens their affinity for actin and actin-related proteins. Second, ERK2 has a tightly defined consensus sequence; however, the proposed phosphorylation sites (and confirmed by us below) do not fit this consensus. We have therefore explored the biological functions of Scar/WAVE phosphorylation in detail. A separate set of phosphorylations is present in the C-terminal VCA domain of Scar/WAVE. It is not detectable by, for example, a change in banding pattern on western blots and is difficult to see by mass spectrometry, so it is far less widely described. We [14] and others [15] have shown that this is constitutive and has a role in tuning the sensitivity of the Scar/WAVE complex rather than activating it; both phosphomimetic and phosphorylation-deficient mutants are active.

One key process in Scar/WAVE biology that is particularly poorly understood is autoactivation. It is clear that pseudopods of migrating cells (which are caused by Scar/WAVE) are controlled through positive feedback—new actin polymerization occurs adjacent to recent pseudopods [8], leading to traveling waves at the edges of cells [16], but the mechanism of this regulation is not well understood [17,18]. It does, however, emphasize the importance of understanding the full dynamics of Scar/WAVE—its recruitment and release from pseudopods, and its synthesis and breakdown—rather than focusing exclusively on its activation.

There is no compelling reason to connect Scar/WAVE phosphorylation to the activation step. Phosphorylation could alter the activity of the complex after it is activated or alter

properties like the rate of autoactivation or the stability of Scar/WAVE once recruited [18]. In the present work, we find that phosphorylation's primary role appears to be centered around biasing pseudopod behavior, as required by pseudopod-based models of cell migration, rather than in initiating new pseudopods or actin polymerization.

## Results

### Detailed analysis of Scar/WAVE phosphorylation in vivo

A number of authors have observed Scar/WAVE phosphorylation [10,11,14, 19,20]. Inactive Scar/WAVE is constitutively phosphorylated in the extreme C-terminus [14]. Additional phosphorylation is seen in (for example) growth factor–stimulated cells, which a number of authors have placed downstream of ERK2 [10,11,19], implying it is part of the process by which signaling causes pseudopods to form. To analyze the phosphorylation in detail, we devised an optimized electrophoresis regime. Examined by western blot on normal SDS-PAGE gels, *Dictyostelium* Scar runs as a single band with a diffuse upper edge; quantitative analysis shows this band forms a single, diffuse peak (Fig 1A). To improve separation of different phosphoforms, we optimized PAGE systems. The best separation occurred on 10% acrylamide gels containing low bis-acrylamide concentrations of 0.06% (low-bis gels) compared with the usual 0.3%. In such gels, Scar from migrating *Dictyostelium* resolves into at least 6 distinct bands (Fig 1B, asterisks) with clear separation on an intensity plot. Lambda phosphatase resolved the multiple bands to a single one, shifting the multiple intensity peaks (solid line) to single peak (dotted line; Fig 1C), confirming that the multiple bands are due to phosphorylation, and Scar is typically hyperphosphorylated to varying degrees.

Similarly, WAVE2 from 2 mammalian lines (human MDA-MB-231 and mouse NIH3T3) appears on western blots as a broad band with a single intensity peak (Fig 1D). As with *Dictyostelium* Scar, low-bis gels reveal multiple phosphorylated WAVE2 bands (Fig 1E, asterisks). Intensity plots show multiple well-resolved peaks that are shifted down after phosphatase treatment confirming the bands represent WAVE2 phosphoforms.

### Identification of phosphorylation sites in vivo

Previous reports identified Scar/WAVE phosphosites using protein expressed in vitro [11] or overexpressed fusion proteins [9]. Because Scar/WAVE's biological roles occur exclusively as part of the 5-membered Scar/WAVE complex, and overexpressed proteins are unevenly and incompletely incorporated into the complex, we analyzed the phosphorylation of native proteins. We purified the *Dictyostelium* Scar complex using *nap1⁻* cells stably transfected with single copies of GFP-Nap1 [14], yielding normal expression levels, and pulled down the complex using GFP-TRAP. Under these conditions, Scar is only purified if incorporated in a properly assembled Scar/WAVE complex. Gel-purified Scar bands (S1A and S1B Fig) were analyzed by liquid chromatography-tandem mass spectrometry (LC-MS/MS) to identify phosphopeptides. Scar/WAVE is composed of N-terminal SH, central B, and polyproline, and C-terminal V, C, and A domains [21]. We identified 3 phosphorylated tyrosines (Y88, 129, 210) in the SH domain, 3 phosphorylated serines (S287, 290, 301) in the polyproline, and 3 (S384, 388, 389) in the V domain (Fig 1F). These are additional to the constitutive phosphorylations found by Ura and colleagues [14] from 430–440 in the A domain. The number of SCAR forms discernible as bands in western blots from migrating cells is therefore not a surprise—at least 16 sites are phosphorylated in vivo in normal cells.

Conspicuously, none of the phosphorylated serines made up MAP kinase consensus sites (S1 Table). This was surprising, as a number of papers report that Scar/WAVE bands

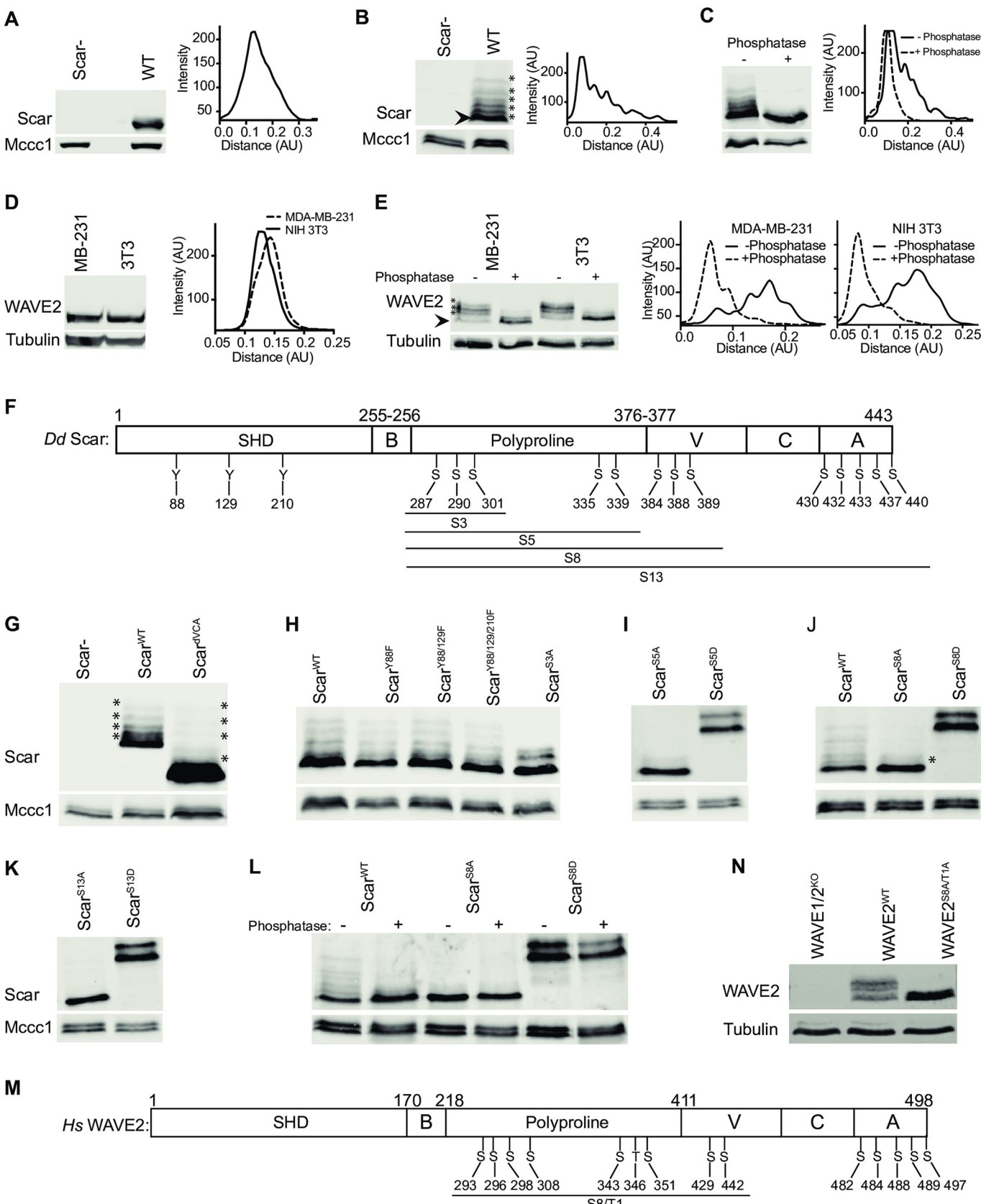

**Fig 1. Multiple phosphorylations in Scar/WAVE. (A)** Western blot of *Dictyostelium* Scar using normal (0.3%) bis-acrylamide gels. Whole cells were boiled in sample buffer and immediately separated on 10% bis-tris gels, blotted, and probed with an anti-Scar antibody, yielding a single diffuse band. Scan of signal intensity shows a single, slightly diffuse peak. Mccc1 is a loading control. **(B)** Western blot of *Dictyostelium* Scar using low (0.06%) bis-acrylamide gels. Whole cells were boiled in sample buffer and immediately separated on 10% bis-tris gels, yielding multiple discrete bands. Scan of signal intensity shows separate peaks. **(C)** Phosphatase treatment. Whole *Dictyostelium* cells were lysed in TN/T buffer, then incubated with and without lambda phosphatase, boiled in sample buffer, and analyzed using low bis-acrylamide gels. The scan shows multiple peaks resolved into one and shifted downwards after phosphatase treatment. **(D)** Western blot of mammalian WAVE2 in mammalian cells using normal (0.3%) bis-acrylamide gels. Whole MDA-MB-231 and NIH 3T3 cells were boiled in sample buffer, separated on 10% bis-tris gels, blotted, and probed with an anti-WAVE2 antibody, yielding a single band. Scan of signal intensity shows a single peak. **(E)** Western blot of mammalian WAVE2 in mammalian cells using low (0.06%) bis-acrylamide gels. Whole MDA-MB-231 and NIH 3T3 cells were boiled in sample buffer and separated on 10% bis-tris gels, yielding at least 5 bands. Incubation with lambda phosphatase resolves the bands into lower bands. Scans of signal intensity show clear multiple peaks resolving incompletely toward 1 lower peak. All experiments were repeated thrice with comparable results. **(F)** Schematic of *Dictyostelium* Scar architecture and identified phosphorylated residues. **(G–K)** Deletions of different amino acids. Lysates from Scar⁻ cells expressing Scar^WT and Scar lacking the VCA domain **(G)**, with Tyr 88, 129, and 201 substituted with Phe **(H)**, Ser 287/290/301 substituted with Ala **(H)**, Ser 287/290/301/335/339 substituted with Ala or Asp **(I)**, Ser 287/290/301/335/339/384/388/389 substituted with Ala or Asp **(J)**, and Ser 287/290/301/335/339/384/388/389/430/432/433/437/440 substituted with Ala or Asp **(K)**, were boiled in sample buffer and analyzed on low (0.06%) bis-acrylamide 10% bis-tris gels. Extra bands in Scar are completely lost after substitution of 13 serines. **(L)** Phosphatase treatment of Scar^WT, Scar^S8A, and Scar^S8D. Lambda phosphatase–treated cell lysates were analyzed using low (0.06%) bis-acrylamide 10% bis-tris gels. The 2 bands of Scar^S8D are barely altered after phosphatase treatment, suggesting the difference is not due to phosphorylation. **(M)** Schematic of human WAVE2 architecture and phosphorylated Ser/Thr residues identified in published and high-throughput screens. **N)** Loss of extra WAVE2 bands in B16F1 mouse cells after substitution of polyproline serines. WAVE1/2KO cells were transfected with intact WAVE2 or WAVE2 with Ser 293/296/298/308/343/351/429/442 and Thr 346 substituted with alanine, then lysed, boiled in sample buffer, separated using low (0.06%) bis-acrylamide 10% bis-tris gels, blotted and probed with an anti-WAVE2 antibody. A, alanine; Ala, alanine; Asp, aspartate; AU, arbitrary unit; D, aspartate; F, phenyl alanine; KO, knock out; Phe, phenyl alanine; S, serine; Ser, serine; T, threonine; Thr, threonine; TN/T, tris-sodium chloride/triton X 100;Tyr, tyrosine; WT, wild type; Y, tyrosine.

(phosphorylation) are uniquely caused by ERK2, which has a strong preference for prolines 2aa N-terminal to and 1aa C-terminal to the target site [22].

To confirm that we had identified the full range of phosphorylations, we tested which amino acids had to be mutated to block extra bands in westerns. We expressed unphosphorylatable (Y-F; S-A) and phosphomimetic (S-D) mutants in *scar*⁻ cells. Removing the 5 phosphorylated serines in the A domain of Scar [14] by deleting the VCA domain (Scar^dVCA) did not reduce the number of bands (Fig 1G; the relative intensities changed but the number did not). Mutation of Y88/129/210F in SHD of Scar did not affect the band shifts (Fig 1H). Mutation of the 3 serines in the polyproline domain (Scar^S3A, Fig 1H) to alanines reduced the number of bands but did not abolish them. Mutating 2 more nearby serines (giving Scar^S5A; S284,290,301,335,339) gave near-complete loss of Scar bands (Fig 1I). Additional mutation of S387/388/389A from the V domain, giving Scar^S8A (Fig 1J), yielded Scar that was nearly homogeneous, with a single very faint band above the main Scar population. The residual heterogeneity was lost in cells whose Scar had also lost the sites in the A domain [14], giving Scar^S13A (Fig 1I, 1J and 1K).

Similarly, mutation of the polyproline serines (Scar^S5D), or the polyproline and V domain serines to aspartate (Scar^S8D), caused a substantial mobility shift and 2 distinct bands on low-bis gel (Fig 1I and 1J). The mobility shift of Scar^S8D was also obvious on a normal-bis gel (S1C Fig), but only a single band of Scar^S8D was resolved. These 2 forms are apparently not caused by phosphorylation; strong phosphatase treatment, which also caused significant proteolysis, caused only partial loss of the upper band (Fig 1L); the difference may be due to a small modifier like ubiquitin, though we were unable to find ubiquitin itself by western blot in any variant (Scar^WT, Scar^S8A or Scar^S8D; S2 Fig).

The mammalian Scar/WAVE2 has been also shown to be phosphorylated at multiple positions (Fig 1M). As with *Dictyostelium*, the sites in human WAVE2 do not fit the MAP kinase consensus (S1 Table), with the exception of T346, which is relatively sparsely phosphorylated (https://phosphosite.org). Again, similarly to *Dictyostelium* Scar, mutation in WAVE2^S8A/T1A (S293/296/298/308/343/351/429/442/T346A) removed all higher-mobility bands (Fig 1N). This confirmed that the multiple bands in WAVE2 are due to serine and threonine phosphorylations.

In conclusion, the multiple bands seen in Scar/WAVE western blots are mostly caused by differential phosphorylation at sites in the polyproline region, with minimal contributions from the V and A domains.

## Scar/WAVE is phosphorylated after activation

The Scar/WAVE complex must interact with the GTP-bound, active form of Rac1 before it can catalyze Arp2/3 complex activation and pseudopod generation [3,23–28]. We would not expect Rac interactions to be important to the phosphorylation if, as described in several publications [3,23,26], the phosphorylation is an upstream process regulating Scar/WAVE activation. We therefore determined the effect of Rac1 binding on Scar/WAVE phosphorylation. First, we inhibited the Rac1 activity in both *Dictyostelium* and B16F1 cells by EHT1864, an effective inhibitor of RacGEF activity [29,30], and assessed the band shifts of Scar/WAVE. EHT1864 treatment resulted in complete loss of Pak-CRIB-mRFPmars2 from cell periphery and depolarized *Dictyostelium* cells (Fig 2A, S1 Video), confirming that the inhibitor greatly reduced active Rac levels. Similarly, lamellipods of B16F1 cells collapsed after EHT1864 treatment; their filamentous actin (F-actin) at cell edges also appeared reduced (red, Fig 2B).

To measure the effect of Rac1 inhibition on Scar phosphorylation, *Dictyostelium* cells were incubated with 50 μM EHT1864 for 2, 5, and 10 minutes, lysed, and analyzed by low-bis western. Inhibition of Rac1 resulted in a strong reduction in Scar phosphorylation (Fig 2C). The intensity profile shows the disappearance of peaks for polyphosphorylated Scar even more clearly (Fig 2D). Quantification of the fraction of Scar in the upper bands shows approximately 50% reduction (Fig 2E). This figure underestimates the total change in phosphorylations on Scar, because the higher bands that are disproportionately lost contain several phosphates. Similarly, inhibition of Rac1 in B16F1 cells by EHT1864 reduced the intensity of the upper bands of WAVE2 within 2 hours; the change became even clearer after 6 hours of treatment (Fig 2F). Intensity plots of WAVE2 bands show strong reduction in the highest peak of WAVE2 (0 hours; pink) after EHT1864 treatment (Fig 2G). Quantification of the ratios of the upper (intense) and lowest bands also suggest a substantial loss of WAVE2 phosphorylation (Fig 2H, again an underestimate of total phosphate loss).

Generalized inhibition of Rac1 activity causes secondary effects, as many Rac-regulated proteins such as kinases (for example PAKs; [31]) are important for other aspects of motility, and inhibitors may have off-target effects. We therefore verified that active Rac1's effect on Scar phosphorylation is direct. We examined cells in which the Pir121 protein was replaced by Pir121-EGFP, with and without the Rac1 nonbinding, A-site mutation [3,5]. Pak-CRIB-mRFP-mars2 [8] localizes to the pseudopods and cell periphery in both strains (Fig 2I and 2J; Panel I, S2 Video), confirming that Rac is activated in wild type (WT) and mutant cells. However, the Scar complex is only localized in cells expressing the functional Pir121EGFP, not the A-site mutant (Fig 2I; Panel II, S2 Video). Scar phosphorylation is greatly diminished in the A-site mutant (Fig 2K), to a level comparable to that seen in cells treated with Rac inhibitor. Taken together, these experiments clearly demonstrate that Scar phosphorylation occurs only after the complex is activated by interaction with Rac1.

To affirm the sequence of activation and phosphorylation, we tested the effect of latrunculin, an actin inhibitor [32,33], which causes exaggerated membrane recruitment of the Scar/ WAVE complex in mammalian cells [34] and *Dictyostelium*. In *Dictyostelium*, latrunculin causes concentrated and persistent recruitment of Scar complex to the membrane and activation of Arp2/3 complex (Fig 2L, S3 Video). Upon latrunculin treatment, Scar phosphorylation rapidly increases; like the Scar recruitment and Arp2/3 activation, the phosphorylation reaches

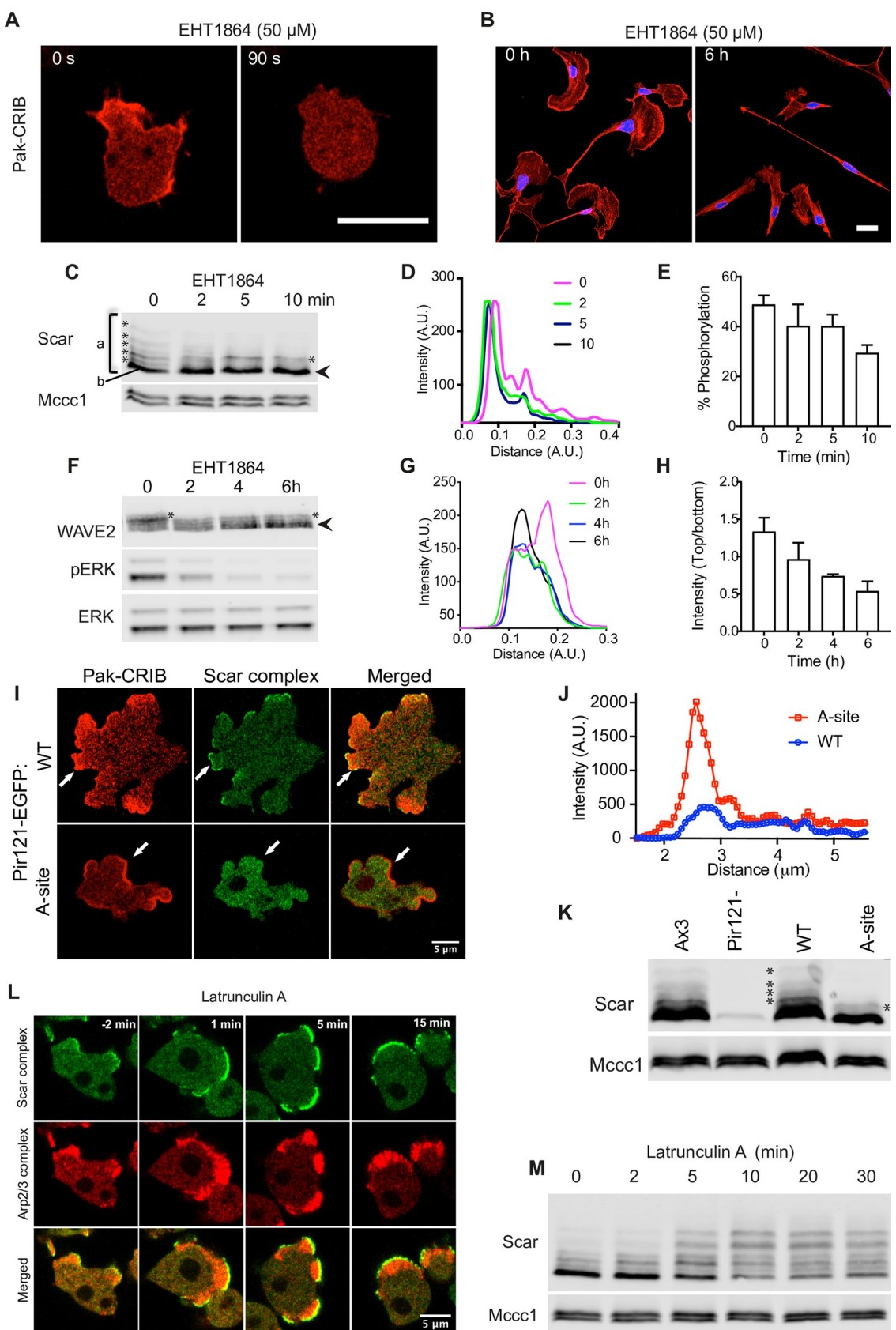

**Fig 2. Rac1 binding and Scar/WAVE phosphorylation. (A)** Effect of Rac1 inhibition by EHT 1864 on Pak-CRIB-mRFPmars2 localization in *Dictyostelium*. *Dictyostelium* cells expressing Rac1 marker, Pak-CRIB-mRFPmars2 were imaged by AiryScan confocal microscopy. Pak-CRIB-mRFPmars2 (Red) rapidly delocalizes after addition of EHT1864 (EHT1864, 50 μM) treatment (Scale bar = 10 μm). **(B)** Effect of Rac1 inhibition by EHT 1864 on lamellipod formation in B16 melanoma cells. B16F1 cells were seeded on Laminin A–coated glass and treated with EHT1864 (50 μM) for 6 hours. Cells were fixed and stained with Alexa fluor 568-phalloidin (red) and DAPI (blue). Untreated cells retained broad lamellipods with intense F-actin at the leading edge, whereas lamellipods and F-actin are compromised in the EHT 1864–treated cells (scale bar = 20 μm). **(C–E)** Effect of Rac1 inhibition by EHT 1864 on *Dictyostelium* Scar phosphorylation. Cells were treated with 50 μM EHT1864 for the indicated times, and lysates were analyzed for Scar band shifts by western blotting using low (0.06%) bis-acrylamide gels. Disappearance of extra bands after EHT 1864 treatment indicates loss of phosphorylated Scar. Graph in **(D)** shows density quantitation of western blots at different times. **(E)** shows ratio of upper (more-phosphorylated, "a"– "b") band to aggregate intensity ("a") of total Scar (bars show mean ± SD, *n* = 3). The numerical data are included in S1 Data. **(F–H)** Effect of Rac1 inhibition by EHT 1864 on B16 melanoma WAVE2 phosphorylation. Cells were treated with 50 μM EHT1864 for the indicated times, and lysates were analyzed for WAVE2 band shifts by western blotting using low (0.06%) bis-acrylamide gels. Disappearance of extra bands after EHT 1864 treatment indicates loss of phosphorylated WAVE2. Graph in **(G)** shows density quantitation of western blots at different times. **(H)** shows the ratio of upper (more-phosphorylated) and lowest (least-phosphorylated) band of WAVE2 (bars show mean ± SD, *n* = 3). The numerical data are included in S1 Data. **(I)** Requirement for Rac1-binding PIR121 A site for Scar complex localization in *Dictyostelium* cells. Pir121⁻ cells expressing Pir121-EGFP with and without mutated A site (K193D/R194D) were coexpressed with Pak-CRIB-RFPmars2 and imaged by confocal microscopy while migrating under agarose up a folate gradient. Upper panel: unaltered PIR121-eGFP; lower panel: Pir121-eGFP$^{A\ site}$. Rac1 localizes (red) to the membrane in both Pir121-EGFP$^{WT}$ and Pir121-EGFP$^{K193D/R194D}$. Scar complex (green) is recruited to sites of Rac1 activation only when Rac can bind. **(J)** Quantitative assessment of Rac1 activation (Pak-CRIB) in WT and A- site mutant from panel I. A line was drawn from the protruding edge through the center of the cell and relative pixel values plotted. **(K)** Requirement for Rac1-binding PIR121 A site for *Dictyostelium* Scar phosphorylation. WT, Pir121⁻, and Pir121⁻ cells expressing Pir121 with and without mutated A site (K193D/R194D) were analyzed for Scar band shifts by western blotting using low (0.06%) bis-acrylamide gels. Phosphorylated Scar bands are absent when PIR121's A site is mutated. **(L)** Localization, distribution, and hyperactivation of Scar complex and Arp2/3 complex after latrunculin A treatment. *Dictyostelium* cells expressing eGFP-NAP1 and mRFP-mars2-ArpC4 were imaged using an AiryScan confocal microscope. 5 μM latrunculin A was added at t = 30 seconds. Scale bar = 10 μm. Scar complex and Arp2/3 recruitment at the cell membrane increase after addition of latrunculin. **(M)** Increased Scar phosphorylation after latrunculin A treatment. *Dictyostelium* cells were treated with latrunculin A for the indicated times, then Scar band shifts were analyzed by western blotting using low (0.06%) bis-acrylamide gels. Phosphorylated bands become markedly more abundant after latrunculin A treatment. All experiments were repeated 3 times. AU, arbitrary unit; WT, wild type.

a far higher level than is normally seen (Fig 2M). Thus, phosphorylation of Scar again correlates with its recruitment and activation, rather than upstream processes.

It is interesting to note that there is currently no biochemical assay for Scar/WAVE activity. The only measures of activation are indirect, for example, recruitment to the extreme leading edge, and downstream consequences, such as actin polymerization. This phosphorylation assay could therefore be a potentially useful indirect measure of recent Scar/WAVE activation in vivo.

## Chemotactic signaling does not control Scar phosphorylation

Earlier reports described the MAP kinase ERK2, induced downstream of chemotactic and growth factor signaling, as the principal driver of Scar/WAVE phosphorylation [9,11,19]. This is important because it provides a direct connection between signaling and pseudopod initiation and growth that is otherwise lacking. As described earlier, we questioned this result, because MAP kinase sites are usually restricted to a consensus that was not seen in the sites we identified (though some flexibility in the consensus has been reported; [35]). We therefore examined the effect of extracellular signals and MAP kinases on Scar/WAVE phosphorylation. Growing *Dictyostelium* cells use folate as a chemoattractant. During multicellular development, they down-regulate folate receptors and express receptors to extracellular cyclic adenosine 3',5'- monophosphate (cAMP). Treatment of growing cells with folate, or developed cells with cAMP, causes increases in polymerized actin levels that are thought to be important for chemotaxis [36,37]. To provide a general narrative, we examined Scar phosphorylation after both treatments. Despite massive activation of erkB protein, no change in the Scar phosphorylation was seen in either folate- or cAMP-treated cells (Fig 3A and 3D), either visually from the gel or when expressed quantitatively as the phosphorylated fraction (Fig 3B, 3C, 3E and 3F).

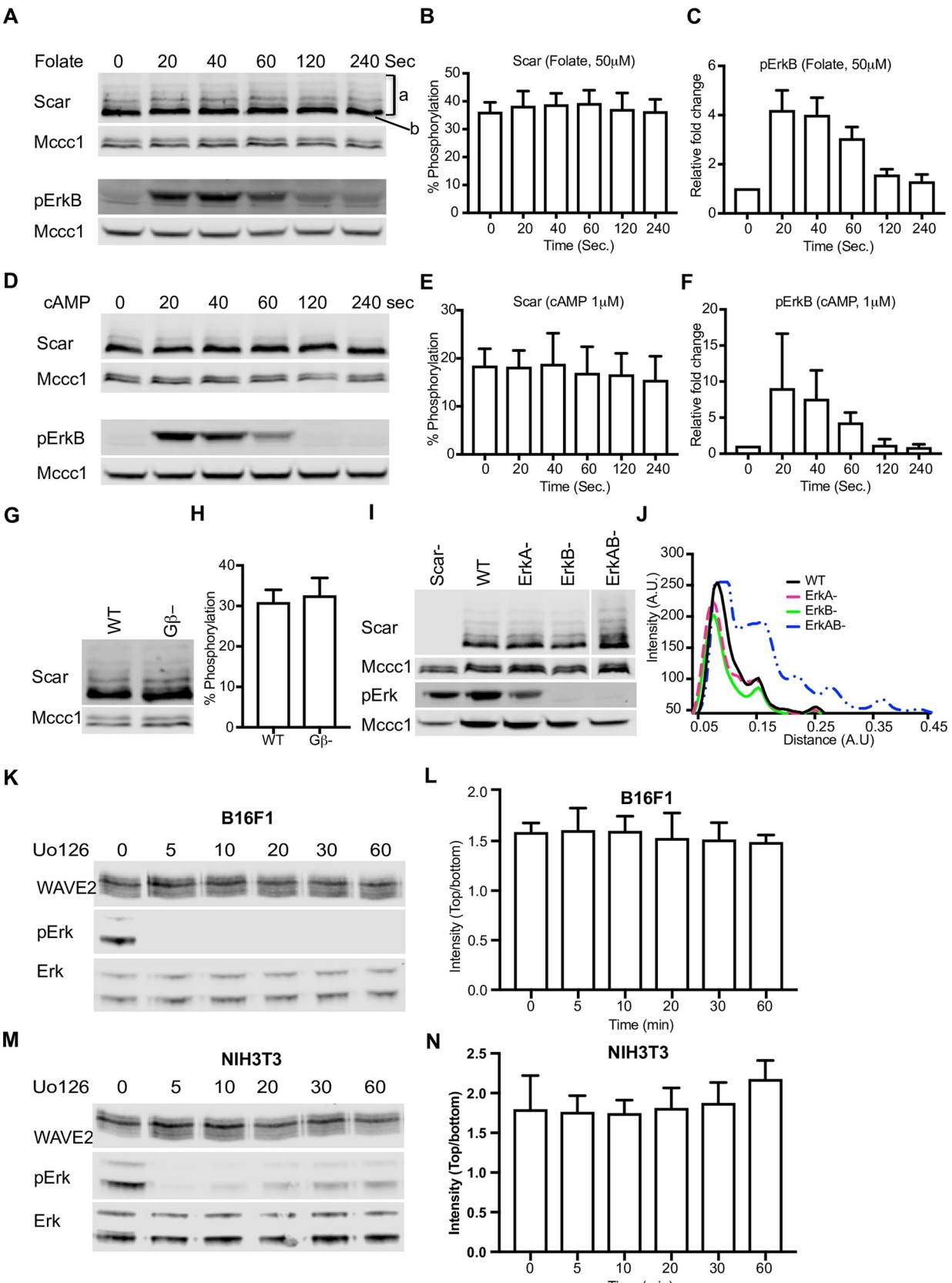

**Fig 3. Signaling and Erk-independent phosphorylation of Scar/WAVE. (A–C)** Folate stimulation and Scar phosphorylation in *Dictyostelium*. Washed growing cells were treated with 50 μM folate for the indicated times, then Scar band shifts were analyzed by western blotting using low (0.06%) bis-acrylamide gels. Folate does not enhance Scar phosphorylation (B; bands show mean ± SD, *n* = 3) despite substantial signaling response shown by erkB phosphorylation (C). The numerical data are included in S1 Data. **(D–F)** cAMP stimulation and Scar phosphorylation in *Dictyostelium*. Aggregation-stage cells were treated with 1μM cAMP for the indicated times, then Scar band shifts were analyzed by western blotting using low (0.06%) bis-acrylamide gels. cAMP does not enhance Scar phosphorylation **(E)** despite very substantial signaling response shown by erkB phosphorylation (bars show mean ± SD; *n* = 3). The numerical data are included in S1 Data. **(G,H)** Scar phosphorylation in WT and Gβ- cells. WT and Gβ- cell band shifts were analyzed by western blotting using low (0.06%) bis-acrylamide gels. Both show similar band shifts of Scar, as reflected in the quantitation of the upper band (H; bars show mean ± SD; *n* = 3). The numerical data are included in S1 Data. **(I)** Scar phosphorylation in MAP kinase (ERK) mutants. Scar⁻, WT, *erk*A⁻, *erk*B⁻ and *erk*AB⁻ cells were analyzed by western blotting using low (0.06%) bis-acrylamide gels. Single mutants show similar phosphorylation; double mutant shows more, as shown by densitometry of the lanes **(J)**. **(K-N)** WAVE2 phosphorylation in B16 **(K,L)** and NIH3T3 **(M,N)** cells after inhibition of Erk activation. Cells were treated with U0126 for the indicated times, then analyzed by western blotting using low (0.06%) bis-acrylamide gels. Erk and phospho-Erk levels were analyzed using normal gels and antibodies against pan-Erk and phospho-Erk. Quantitation of the phosphorylated bands shows no change despite essentially complete inhibition of Erk (bars show mean ± SD, *n* = 3). The numerical data are included in S1 Data. cAMP, cyclic 3',5'-adenosine monophosphate; ERK, extracellular signal regulated kinase; Gβ, G-protein beta subunit; WT, wild type.

Starved cells have less phosphorylated Scar than vegetative ones, so both basal and stimulated levels are lower (S3 Fig). We do not know whether this is caused by decreased phosphorylation, increased dephosphorylation, or decreased stability of the phospho-Scar.

Knockouts of the *Dictyostelium* Gβ protein show a complete loss of G-protein signaling [38] and reduced ErkB activity [39]; because there are also no tyrosine kinase receptors in *Dictyostelium*, these cells are complete chemotactic nulls. However, Gβ⁻ mutant cells have normal levels of Scar phosphorylation (Fig 3G and 3H), as well as morphologically normal pseudopods [40]. This confirms that chemotactic signaling is not required for Scar phosphorylation.

We determined the importance of MAP kinase signaling for Scar/WAVE phosphorylation in both *Dictyostelium* and mammalian cells. First, we examined the band shifts for Scar in western blots from *erk*A⁻, *erk*B⁻, and *erk*AB⁻ nulls of *Dictyostelium* [41]. The band shifts on the western blot and quantitated peaks were not discernibly different from the WT parent (Fig 3I and 3J). Because ErkA and ErkB are the only MAP kinases in *Dictyostelium*, MAP kinase signaling is not important for Scar phosphorylation.

To confirm this result is general, we examined WAVE2 phosphorylation in mammalian NIH3T3 and B16F1 cells after inhibition of Erk2 activity. Treatment with U0126, an efficient inhibitor of MEK [42] immediately abolished Erk2 phosphorylation (and thus activity) in both NIH3T3 and B16F1 cells but did not change the phosphorylation state of WAVE2 even after prolonged incubation (Fig 3K, 3L, 3M and 3N). It is of course possible that Erk2 phosphorylates the Scar/WAVE complex under some conditions, but these results confirm that the Scar/WAVE phosphorylation we observe is neither executed by Erk2 nor requires Erk2 function.

## Physical adhesion promotes Scar/WAVE phosphorylation

More generally, our data suggest Scar/WAVE phosphorylation is not primarily regulated by extracellular signaling. Another regulator of pseudopods and lamellipods is cell–substrate adhesion. We tested its role using *Dictyostelium*'s ability to grow either in suspension or adhesion. First, we compared Scar phosphorylation in cells maintained in suspension and adherent cultures. More highly phosphorylated bands (* in Fig 4A) are more intense in cells grown adhering to petri plates, when compared with suspension-grown cells. Intensity plots clearly show more peaks and greater intensities of more-phosphorylated peaks in adherent cells (Fig 4B). This suggests that adhesion stimulates Scar phosphorylation. To investigate this in more detail, we allowed suspensions of cells to adhere to a plastic surface, revealing a clear and significant (*p* = 0.0089; Kruskal–Wallis test) up-regulation of Scar phosphorylation (Fig 4C and 4D). The proportion of Scar in phosphorylated bands increases from 25 ± 6.9% to 42.5 ± 3.5%

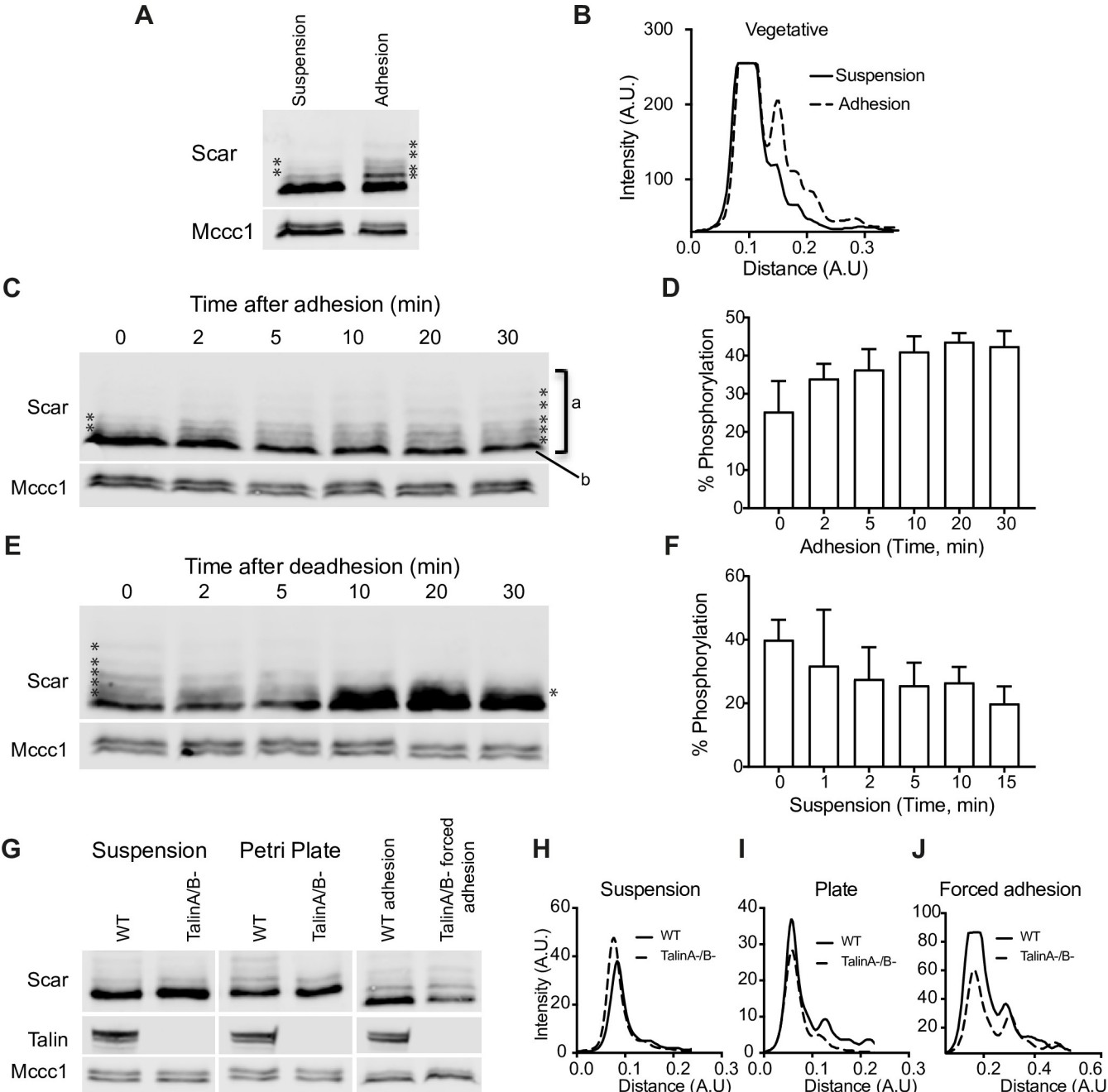

**Fig 4. Physical adhesion enhances Scar/WAVE phosphorylation. (A,B)** Wild-type *Dictyostelium* cells were grown in shaken suspension or adhering to Petri dishes, then analyzed by western blotting using low (0.06%) bis-acrylamide gels. Adherent cells have more and intense bands of Scar/WAVE. Densitometry of total lane intensities shows a large difference in the phosphorylated Scar bands. **(C,D)** Acute effect of adhesion on Scar/WAVE phosphorylation. Cells growing in suspension were allowed to adhere in petri dishes and lysed at indicated time points. Cell lysates were analyzed by western blotting using low (0.06%) bis-acrylamide gels. The number and intensity of bands (*) increases with time of adhesion. Quantitation of lane intensities (intensity of "a"− "b"/total intensity of "a"; mean ± SD, *n* = 3) confirms a progressive increase in the intensities of phosphorylated Scar bands. The numerical data are included in S1 Data. **(E,F)** Acute effect of de-adhesion on Scar/WAVE phosphorylation. Cells grown in Petri dishes were dislodged by a jet of medium from a P1000 pipette and maintained in suspension (2×10⁶ cells/ml) by shaking at 120 rpm. Cell lysates were analyzed by western blotting using low (0.06%) bis-acrylamide gels. Quantitation of lane intensities (mean ± SD, *n* = 3) confirms a progressive loss in intensity of phosphorylated Scar bands. The numerical data are included in S1 Data. **(G–J)** Relative requirement of Talin and adhesion. WT and *talA⁻/ talB⁻* cells were allowed to make physical contact with plastic Petri dishes and where appropriate, forced to adhere by withdrawing the liquid with 3MM paper. Cell lysates were analyzed by western blotting using low (0.06%) bis-acrylamide gels. Densitometry of lane intensities shows a large difference between WT and mutant cells on Petri dishes **(I)** but not when adhesion was forced **(J)**. WT, wild type.

(mean ± SD; Fig 4D). Conversely, when we de-adhered cells using a stream of buffer from a pipette, we saw an obvious reduction in phosphorylated bands (Fig 4E). The proportion of Scar in phosphorylated bands decreased from 40 ± 5 to 20 ± 4 (mean ± SD; Fig 4F). Thus, cell–substrate adhesion induces Scar phosphorylation.

We examined whether cell–substrate adhesion coupled to phospho-Scar via adhesion-linked signaling or through a physical process such as mechanical deformation. This was enabled by talin A/B double-knockout *Dictyostelium* cells. Talin is a key link between adhesion molecules like integrins (and the functionally conserved SibA family in *Dictyostelium*) and the actin cytoskeleton [43,44]. In the absence of Talin, outside-in signaling from integrins to cytoplasm does not occur. *tal*A/B⁻ mutant cells completely fail to adhere when they encounter a substrate [45] and consequently remain spherical and pseudopod-free when cultured, even in dishes where normal cells adhere. This is mirrored by failure to phosphorylate Scar in response to adhesion—in *tal*A/B⁻ mutant cells, the phosphorylated bands were unchanged when cells were allowed to settle in a petri dish (Fig 4G, 4H and 4I).

However, we were able to force *tal*A/B⁻ cells into adhesion to plastic, glass, or 1% agar surfaces by aspirating the liquid so cells were compressed by capillary forces. This yields cells that are mechanically coupled to the substrate without talin-based signaling. Under these conditions, Scar phosphorylation increases in *tal*A/B⁻ just as it does in WT cells (Fig 4G and 4J). This implies that Scar/WAVE phosphorylation is mediated by the physical process of attachment—through mechanochemical processes, for example—rather than through canonical integrin adhesion signaling.

## Phosphorylation-deficient and phosphomimetic Scar mutants are fully functional

In essentially every lamellipod or pseudopod of normal cells, actin and Arp2/3 complex recruitment are mediated by the activation of the Scar/WAVE complex [2]. Phosphorylation of the polyproline domain has been described as important for Scar/WAVE complex activation and formation of such protrusions [9–11,14,19,20]. To test the importance of polyproline phosphorylation in vivo, we constructed mutants that were unphosphorylatable (Scar^S8A) and phosphomimetic (Scar^S8D) at the serines we had identified (Fig 1F and 1J) and expressed them in *scar*⁻ cells. Both mutants were expressed at the same levels as unmutated Scar (Fig 1J) and at similar levels to untransfected parental cells (S4A and S4B Fig). Westerns of Pir121, Nap1, Scar, and Abi from immunoprecipitates showed that Scar^WT, Scar^S8A, and Scar^S8D were all found in complete, stable complexes (S4A and S4C Fig); phosphorylation of Scar was also identical in rescued and untransfected cells (S4D and S4E Fig).

Cells lacking Scar migrate inefficiently and are less directional. They migrate mainly by blebbing[8], and when pseudopods are made, they are small (Fig 5A; Panel I, S4 Video). In contrast, *scar*⁻ cells rescued with Scar^WT, Scar^S8A, and Scar^S8D formed longer pseudopods that moved faster and split frequently (Fig 5A; S4 Video). The migration speed of knockout cells was completely rescued by expression of wild-type or either phosphomutant Scar (Fig 5B), with phosphorylation changes causing a slight increase in cell speed. The tortuosity of the cell perimeters, which describes the amount of recent protrusion and shape change, was low in *scar*⁻ cells but rescued by Scar^S8A and Scar^S8D just as well as by Scar^WT (Fig 5C). Thus, phosphorylation is clearly neither a precondition for activity nor a direct cause of SCAR inactivation. However, the dynamics of pseudopods were obviously altered (Fig 5D; S4 Video). Scar⁻ cell pseudopods lasted for a very short duration (7.4 ± 2.5 seconds, mean ± SD, compared with WT 16 ± 5.7 seconds). Pseudopods in Scar^S8A-expressing cells extended for far longer (37.1 ± 24.6 seconds). Scar^S8A pseudopods were also larger and broader (Fig 5E and 5F).

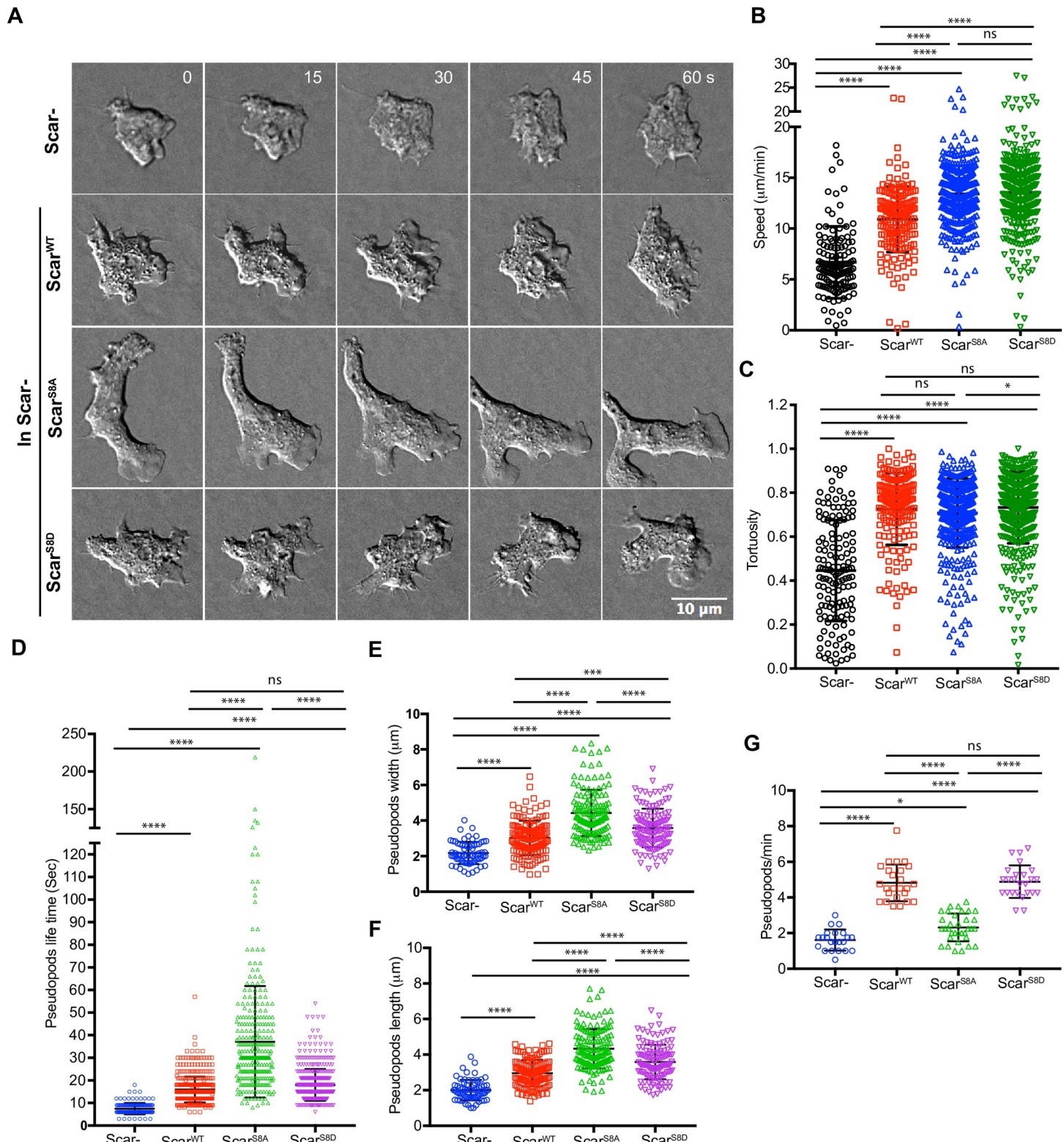

**Fig 5. Scar/WAVE phosphorylation mutants are functional. (A–C)** Rescue of pseudopod formation by mutated Scar. Scar- cells were transfected with Scar$^{WT}$, Scar$^{S8A}$, and Scar$^{S8D}$ and allowed to migrate under agarose up a folate gradient while being observed by DIC microscopy at a frame interval of 3 seconds (1f/3s). All rescued cells formed actin pseudopods; those formed by Scar$^{S8A}$-expressing cells were broad and elongated. Panel **(B)** shows quantitation of speeds and **(C)** the tortuosity (total distance migrated/net distance migrated) of paths in **(B)** (mean ± SD; $n = 140^{Scar-}\ 170^{WT}$, $413^{S8A}$, $437^{S8D}$ over 3 independent experiments; $^{****}p \leq 0.0001$, $^{*}p \leq 0.05$, $^{ns}p \geq 0.05$,

1-way ANOVA, Dunn's multiple comparison test). The numerical data are included in S1 Data. **(D–G)** Effects of Scar$^{WT}$, Scar$^{S8A}$, and Scar$^{S8D}$ on pseudopod dynamics. Pseudopod lifetime was measured from DIC videos. Pseudopod length and width were measured using ImageJ from single frames, and generation rate calculated from the number of pseudopods lasting at least 2 frames. Scar$^{S8A}$ yields increased, and Scar$^{S8D}$ normal, lifetimes. Scar$^{S8A}$ pseudopods are longer-lived than Scar$^{S8D}$ and Scar$^{WT}$, and Scar$^{S8A}$ pseudopods generated less frequently than Scar$^{WT}$ and Scar$^{S8D}$ (mean ± SD; $n > 25$ cells; $^*p \leq 0.05$, $^{**}p \leq 0.01$, $^{****}p \leq 0.0001$, 1-way ANOVA, Dunn's multiple comparison test). The numerical data are included in S1 Data. DIC, differential interference contrast; ns, not significant; WT, wild type.

Because each pseudopod had a greater size and lasted longer, they were also formed less often (Fig 5G). In contrast, Scar$^{S8D}$-driven pseudopods had normal lifetimes and were made at same rate as Scar$^{WT}$.

## Phosphorylation-deficient Scar is recruited in larger and longer-lived patches

Active Scar complex localizes to the fronts of cells and causes actin protrusions [8]. Tagging Scar itself with GFP greatly alters both Scar and pseudopod dynamics. To observe the effects of Scar phosphorylation under physiological conditions, we therefore expressed Scar mutants in Scar- cells in which Nap1 has been replaced by a single copy of GFP-Nap1 (i.e., scar$^-$/Nap1$^{GFP}$ cells; [14]). Scar$^{WT}$, Scar$^{S8A}$, and Scar$^{S8D}$ all localized efficiently at the pseudopod periphery, but there were marked differences in dynamics (Fig 6A; S5 Video). As usual [14], Scar$^{WT}$ was recruited in localized bursts of fairly short duration ("Scar patches"; 8.2 ± 5.9 seconds, mean ± SD). The phosphorylation-deficient Scar (Scar$^{S8A}$) remained recruited for a substantially longer duration (13.7 ± 11.63 seconds, mean ± SD). As well as being longer-lived (Fig 6B), Scar$^{S8A}$ patches were also larger (Fig 6C). This led to changes in patch frequency–the larger, longer-lived Scar$^{S8A}$ patches were made at a lower frequency than Scar$^{WT}$ (Fig 6D). Conversely, localization of Scar$^{S8D}$ was even briefer (6.37 ± 1.94 seconds, mean ± SD), and patches were made at an even higher frequency than Scar$^{WT}$, though they were about the same size.

From these observations, we hypothesized that Scar/WAVE phosphorylation is not required for activation but rather modulates cell migration by controlling the dynamics of Scar/WAVE patches. To confirm this, we constructed nearly completely unphosphorylated and phosphomimetic Scar by additionally mutating the 5 serine residues of the acidic domain [14]. Scar complex and F-actin were both beautifully accumulated in the pseudopods of cells expressing Scar$^{S13A}$ and Scar$^{S13D}$ and were similar to Scar$^{WT}$ (S6 Video).

To confirm this observation in mammalian cells, we examined the effect of phosphorylation-deficient mutations (WAVE2$^{S8A/T1A}$; Fig 1N) in B16F1 cells. WAVE1/2 KO cells transfected with empty vector did not form lamellipods (Fig 6E; S7 Video). However, expressing either WAVE2$^{WT}$ and WAVE2$^{S8A/T1A}$ in knockout cells rescued lamellipod formation (Fig 6E and 6F). Quantitative examination revealed that WAVE2$^{S8A/T1A}$ driven lamellipods were longer-lived than WAVE2$^{WT}$ driven ones (Fig 6G). Lamellipod width and cell perimeter of WAVE2$^{S8A/T1A}$ cells were also significantly increased (Fig 6H and 6I). WAVE2$^{WT}$ and WAVE2$^{S8A/T1A}$ equally rescued the random migration velocity of WAVE1/2 KO (Fig 6J; 1.3 ± 0.4 μm/minutes and 1.1 ± 0.2 μm/minute versus 0.5 ± 0.1μm/minute; mean ± SD). These results clearly show that Scar/WAVE does not require phosphorylation to be functional.

## Scar phosphomutants rescue scar-/wasp- cells

*Dictyostelium* cells lacking Scar can repurpose Wiskott-Aldrich Syndrome protein (WASP) to drive pseudopod formation and Arp2/3 complex activation [8,46], which complicates the phenotypes of *scar* mutants. *Scar$^-$/wasp$^-$* cells cannot move and cannot grow; we have therefore developed an inducible double-knockout (*scar$^{tet}$/wasp$^-$*) cell line, in which Scar expression

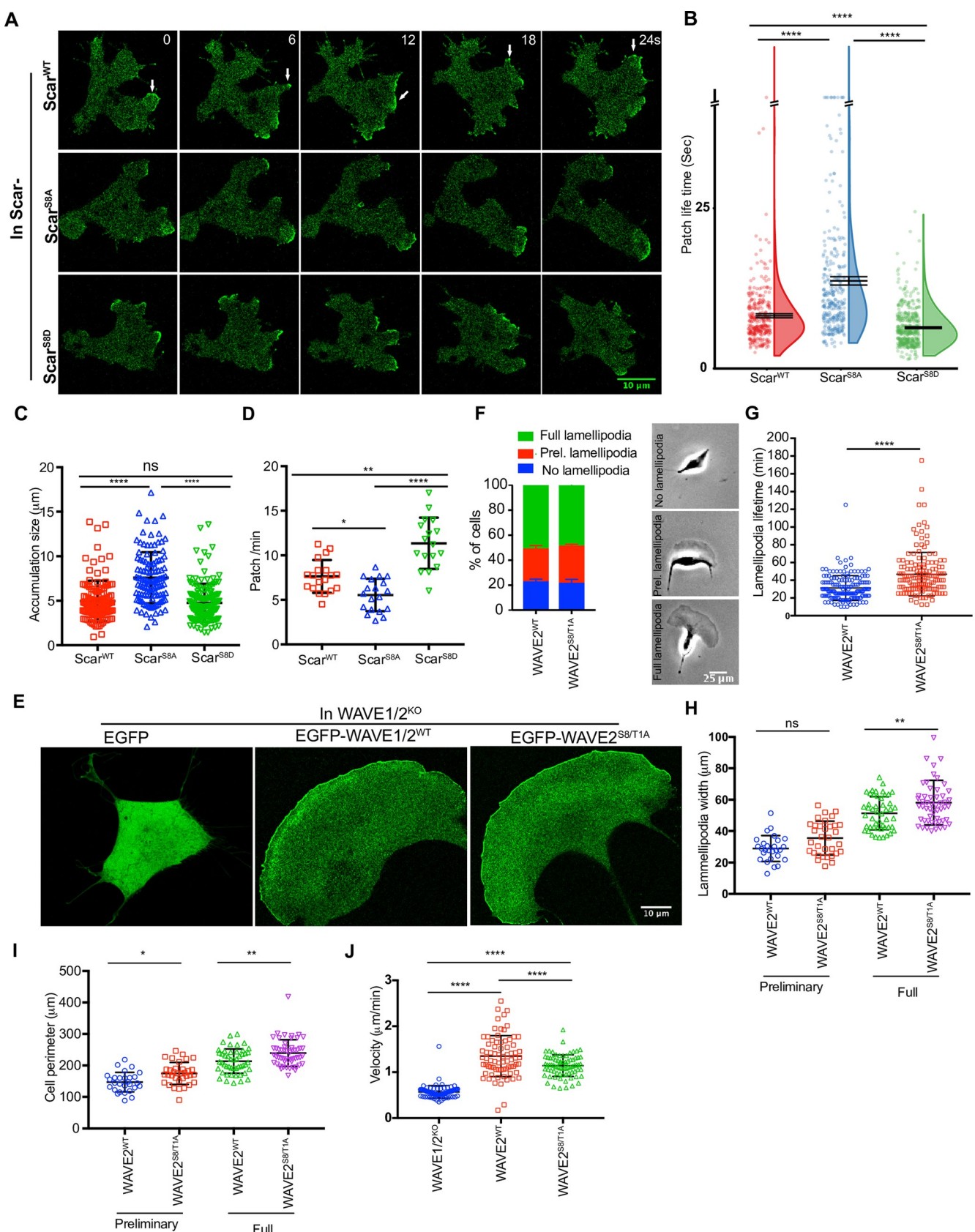

**Fig 6. Scar/WAVE phosphorylation mutants are recruited to pseudopods and lamellipods. (A)** Subcellular localization of Scar complex. Scar$^{WT}$, Scar$^{S8A}$, and Scar$^{S8D}$ were expressed in Scar$^-$/Nap1$^-$/eGFP-NAP1 cells, allowed to migrate under agarose up a folate gradient and examined by AiryScan confocal microscopy. All show efficient Scar/WAVE complex (green) localization to the pseudopods. **(B, C, D)** Lifetimes, size, and generation rates of mutant Scar patches (arrows in panel A). Scar$^-$/Nap1$^-$/eGFP-NAP1 cells expressing Scar$^{WT}$, Scar$^{S8A}$, and Scar$^{S8D}$ were allowed to migrate up folate gradients under agarose, EGFP localization was observed by AiryScan confocal microscopy at a frame interval of 3 seconds (1f/3s). Patch lifetime was measured from the number of frames a patch showed continuous presence of labeled Scar complex, size was measured in ImageJ from individual frames, and generation rate calculated from the number of patches lasting at least 2 frames. Scar$^{S8A}$ shows increased and Scar$^{S8D}$ diminished lifetime compared with Scar$^{WT}$. Scar$^{S8A}$ patches are longer-lived than Scar$^{S8D}$ and Scar$^{WT}$, and Scar$^{S8A}$ generates less and Scar$^{S8D}$ more frequently than Scar$^{WT}$ (mean ± SD; $n > 25$ cells; $^*p \leq 0.05$, $^{**}p \leq 0.01$, $^{****}p \leq 0.0001$, 1-way ANOVA, Dunn's multiple comparison test). The numerical data are included in S1 Data. **(E)** Rescue of lamellipod formation by WAVE2 phosphomutants in melanoma cells. B16F1-WAVE1$^-$/2$^-$ cells expressing EGFP-WAVE2$^{WT}$ and EGFP-WAVE2$^{S8A/T1A}$ were allowed to adhere on laminin A–coated coverslips and observed by AiryScan confocal microscopy at a frame interval of 20 seconds (1f/20s). WAVE2$^{WT}$ and WAVE2$^{S8A/T1A}$ both rescue lamellipod formation. **(F)** Quantification of lamellipod formation in WAVE1/2 KO cells rescued with EGFP-WAVE2$^{WT}$ and EGFP-WAVE2$^{S8A/T1A}$. Cells were plated on laminin-coated 6-well plates and imaged at 10× by phase contrast time-lapse (1 frame/2.5 minutes). Cells without lamellipods, forming preliminary lamellipods, and full lamellipods were quantified. **(G)** Lamellipod lifetime was measured directly from time-lapse videos. WAVE2$^{S8A/T1A}$ lamellipods last longer. **(H,I)** Increased lamellipod width and cell perimeter. Velocity was calculated using the Image J chemotaxis tool. (mean ± SD; $n > 75$ cells; $^*p \leq 0.05$, $^{**}p \leq 0.01$, $^{****}p \leq 0.0001$, 1-way ANOVA, Dunn's multiple comparison test). The numerical data are included in S1 Data. **(J)** Random migration following rescue with WAVE2 or phosphorylation-deficient WAVE2$^{S8A/T1A}$. Loss of phosphorylation does not block WAVE2's ability to rescue migration. EGFP, enhanced green fluorescent protein; KO, knock out; WT, wild type.

depends on the presence of doxycycline, to test Scar function without WASP complementation [46]. We exploited this line to test Scar phosphomutants' ability to support growth and pseudopod formation. *Scar*$^{tet}$/*wasp*$^-$ cells were transfected with Scar$^{S8A}$ and Scar$^{S8D}$ then deprived of doxycycline 48 hours prior to experiments to remove native Scar. As expected [47], protein levels of expressed Scar$^{WT}$ and both mutants are very similar to those in untransfected wild-type cells (Fig 7A). Expression of both Scar$^{S8A}$ and Scar$^{S8D}$ is dominant; expression of the native Scar is suppressed even in the presence of doxycycline (Fig 7A).

Both Scar$^{S8A}$ and Scar$^{S8D}$ fully rescued the growth of *Scar*$^{tet}$/*wasp*$^-$ cells (Fig 7B). To study the migratory phenotype, cells were examined migrating under agarose up a folate gradient. As previously observed [46], repressed *Scar*$^{tet}$/*wasp*$^-$ cells made long, stable filopods but could not form pseudopods or migrate. However, cells expressing either Scar$^{S8A}$ or Scar$^{S8D}$ showed clear localization of Scar complex to protrusions and F-actin polymerization (Fig 7C and 7D; S8 Video), and robust pseudopod formation (Fig 7E; S9 Video). Both mutants rescued cell migration effectively (*scar*$^-$/*wasp*$^-$ cells do not migrate under the agar, so their speed cannot be measured in this assay, but it is essentially zero). Scar$^{S8A}$ supported a substantially higher speed (8.35 ± 3.2 μm/minute, mean ± SD) than either WT (3.71 ± 2 μm/minute, mean ± SD) or Scar$^{S8D}$ (5.2 ± 2.1 μm/minute, mean ± SD; Fig 7F).

Thus, phosphorylation-deficient and phosphomimetic mutants are fully functional for growth and cell migration, without requiring cooperation from WASP.

## sepA kinase and scar phosphorylation

Because the Scar/WAVE phosphorylation we observe is not mediated by Erk2, we sought other kinases. We identified kinases that interact with the Scar complex by lightly cross-linking EGFP-Nap1 to its neighbors using formaldehyde (the strategy that enabled Fort, Batista and colleagues to identify the Scar/WAVE competitor CYFIP-related Rac1 interactor (CYRI); [4], followed by immunoprecipitation of GFP and mass spectrometric identification of bound proteins. This revealed sepA (DDB_G0276465) as a strong candidate. sepA was originally named "septase" because it is important for normal cytokinesis [48]; it was also identified as a "regulator of adhesion and motility" by Lampert [49] because mutant cells showed excessive adhesion. Both observed phenotypes are consistent with known Scar complex roles. SepA is a ste20 family member with a number of mammalian homologues, notably, MAP3K19 and MAP3K3.

Scar phosphorylation was substantially diminished in sepA$^-$ cells (Fig 8A). The multiple high-motility peaks on low-bis gels were lost (Fig 8B). SepA- cells, unlike WT, did not show

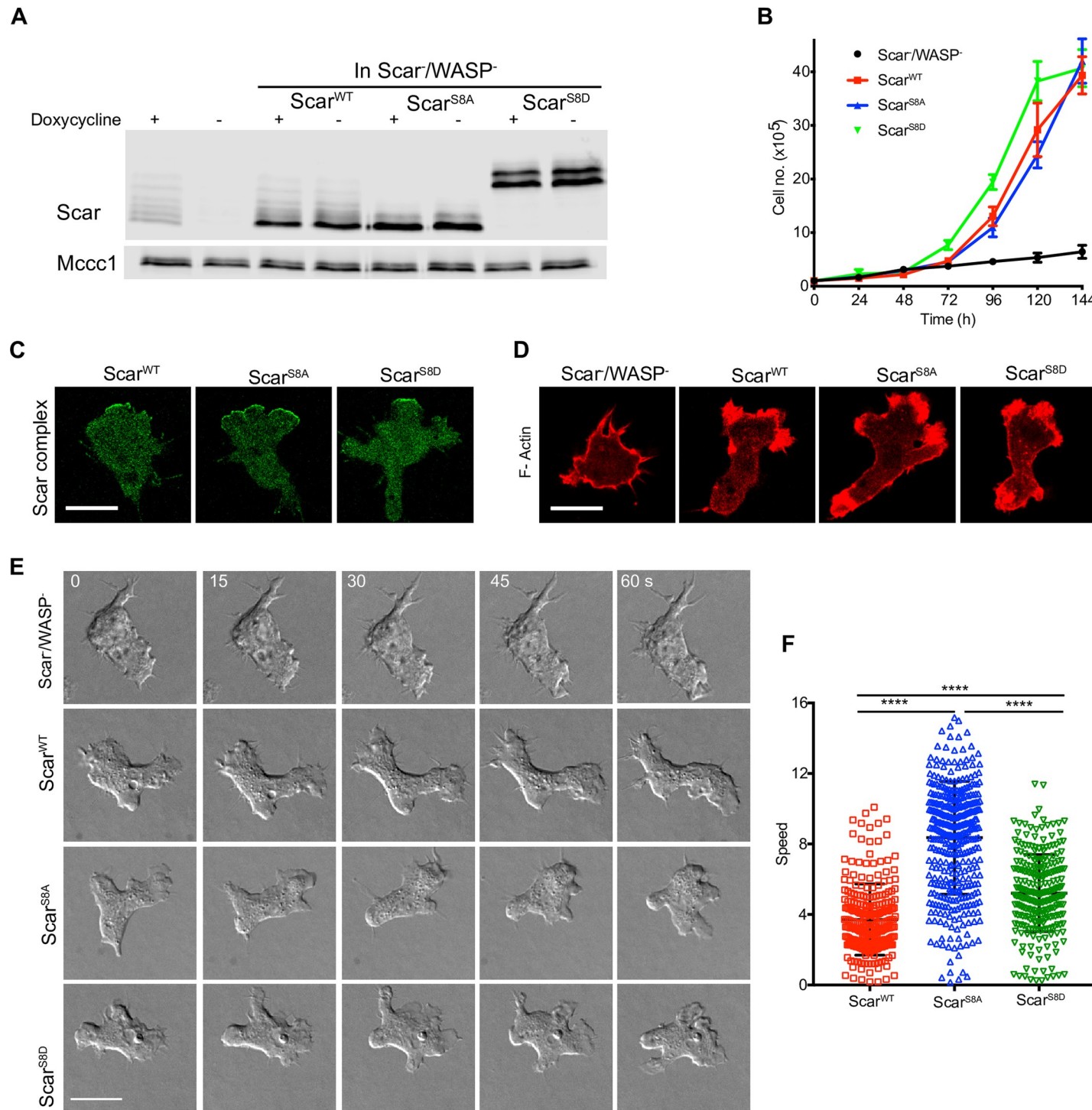

**Fig 7. Scar phosphomutants rescue growth and pseudopods in *scar⁻*/*wasp⁻* cells.** (A) Expression of Scar^WT, Scar^S8A, and Scar^S8D in cells lacking endogenous Scar and WASP. scar^tet/wasp⁻ cells expressing Scar^WT, Scar^S8A, and Scar^S8D were grown with or without doxycycline and analyzed by western blotting using low (0.06%) bis-acrylamide gels. MCCC1 was used as a loading control. **(B)** Rescue of Scar/WASP mutant cell growth. scar^tet/wasp⁻ cells were transfected with Scar^WT, Scar^S8A, and Scar^S8D, expression of endogenous Scar was blocked by removing doxycycline at t = 0, and cell growth in suspension was measured using a Casy cell counter (Innovatis). Points show mean ± SEM, *n* = 3. The numerical data are included in S1 Data. **(C)** Localization of Scar complex in scar^tet/wasp⁻ cells expressing Scar phosphomutants. Scar^WT, Scar^S8A, and Scar^S8D were coexpressed with HSPC300-EGFP in Scar^tet/WASP⁻ cells, and cells migrating under agarose up a folate gradient were imaged by AiryScan confocal microscopy. Scale bar = 5 μm. **D)** Localization of F-actin in scar^tet/wasp⁻ cells expressing Scar phosphomutants. Scar^WT, Scar^S8A, and Scar^S8D were coexpressed with LifeAct-mRFPmars2 in scar^tet/wasp⁻, grown with doxycycline to maintain normal Scar, then kept without doxycycline for 48 hours so only the mutant was expressed. In cells without Scar or WASP, LifeAct was observed in thin protrusions, but the pseudopods of Scar^WT-, Scar^S8A-, and Scar^S8D-expressing cells contained

normal F-actin levels. Scale bar = 5 μm. (**E**) Rescue of pseudopod formation by mutated Scar. scar$^{tet}$/wasp$^-$ cells were transfected with Scar$^{WT}$, Scar$^{S8A}$, and Scar$^{S8D}$, and allowed to migrate under agarose up a folate gradient while being observed by DIC microscopy. All rescued cells formed actin pseudopods; those formed by Scar$^{S8A}$-expressing cells were broad and elongated. Scar$^{WT}$-, Scar$^{S8A}$-, and Scar$^{S8D}$-expressing cells formed pseudopods. Scale bar = 5 μm. (**F**) Quantitation of speeds observed by 10× phase contrast microscopy. Almost no unrescued scar$^{tet}$/wasp$^-$ cells migrated under the agar, so no speed could be measured. (mean ± SD; $n > 278$ cells from 3 independent experiments; $****p \leq 0.0001$, 1-way ANOVA, Dunn's multiple comparison test). The numerical data are included in S1 Data. DIC, differential interference contrast; F-actin, filamentous actin; WASP, Wiskott-Aldrich Syndrome protein; WT, wild type.

stimulation of Scar phosphorylation when allowed to adhere (Fig 8C and 8D). Thus, sepA is an important contributor to Scar phosphorylation.

To confirm the physiological relationship between sepA and Scar phosphorylation, we explored pseudopod and Scar complex recruitment dynamics in mutant cells. SepA- cells formed pseudopods that were very similar to Scar$^{S8A}$-expressing cells. They made exaggeratedly long pseudopods, which were also long-lived (40.1 ± 34.1 seconds versus 12.8 ± 4.3 seconds mean ± SD; Fig 8E and 8F, S10 Video). Being longer-lived, they were also made less frequently (Fig 8G). We also examined recruitment of the Scar complex in sepA- cells using HSPC300-GFP [8]. Scar complex patches were broader, less frequent, and longer-lived in sepA- cells (27.7 ± 29.5 seconds versus 9 ± 3.2 s, mean ± SD; S11 Video, Fig 8H, 8I and 8J). The similarity between the behavior of Scar in sepA- cells and Scar$^{S8A}$ supports and confirms the hypothesis that sepA is an important contributor to Scar phosphorylation.

## Discussion

We have used an improved assay for phosphorylation of Scar/WAVE's polyproline domain to open a window on pseudopod dynamics. This has revealed a number of surprises. Phosphorylation occurs on multiple sites after Scar/WAVE is activated by proteins such as Rac-GTP and reveals an adhesion-dependent, signaling-independent pathway. It is clear that these phosphorylations are not important for Scar/WAVE to be active, and it seems likely that they do not inhibit its activation either. Rather, phosphorylation appears to be a subtle but important tool by which cells can manipulate the dynamics (both recruitment and loss) of Scar/WAVE patches and thus pseudopods. Pseudopod lifetime intersects with a wide range of cell biology —it underlies the mechanism of chemotactic steering [50,51] and regulates cell polarity, persistence [52], and shape change [53]—and mutants with uncontrolled pseudopod lifetime have serious motility defects [4]. Thus, although the function of Scar/WAVE phosphorylation is different from what has been believed, it is fundamentally physiologically important.

Our finding that signaling does not greatly alter the rate of Scar/WAVE phosphorylation (and thus, activation) does not imply that signaling is not important to pseudopod formation and evolution. Chemotaxis—cells migrating up gradients of soluble signaling molecules—clearly happens and is clearly mediated by pseudopods. However, we [50] and others [51] have found that normal chemotaxis rarely involves initiation of new pseudopods—rather, pseudopod generation is mostly random, but localized receptor activation modulate the positions where pseudopods evolve, their lifetime and stability after they are formed, and sometimes the pattern in which new split pseudopods are directed. None of these measured processes require extracellular signals causing direct activation of pseudopod catalysts; rather, more subtle changes like the precise subcellular localization and timing of Scar/WAVE activation could be biased by chemoattractant signaling. Equally, chemoattractants could work through pathways other than Scar/WAVE activation, biasing the growth and lifetimes of pseudopods rather than their initiation. This is in full agreement with the pseudopod-centered view of chemotaxis [54], which emphasizes that pseudopods are self-organized, rather than the outcome of external signals.

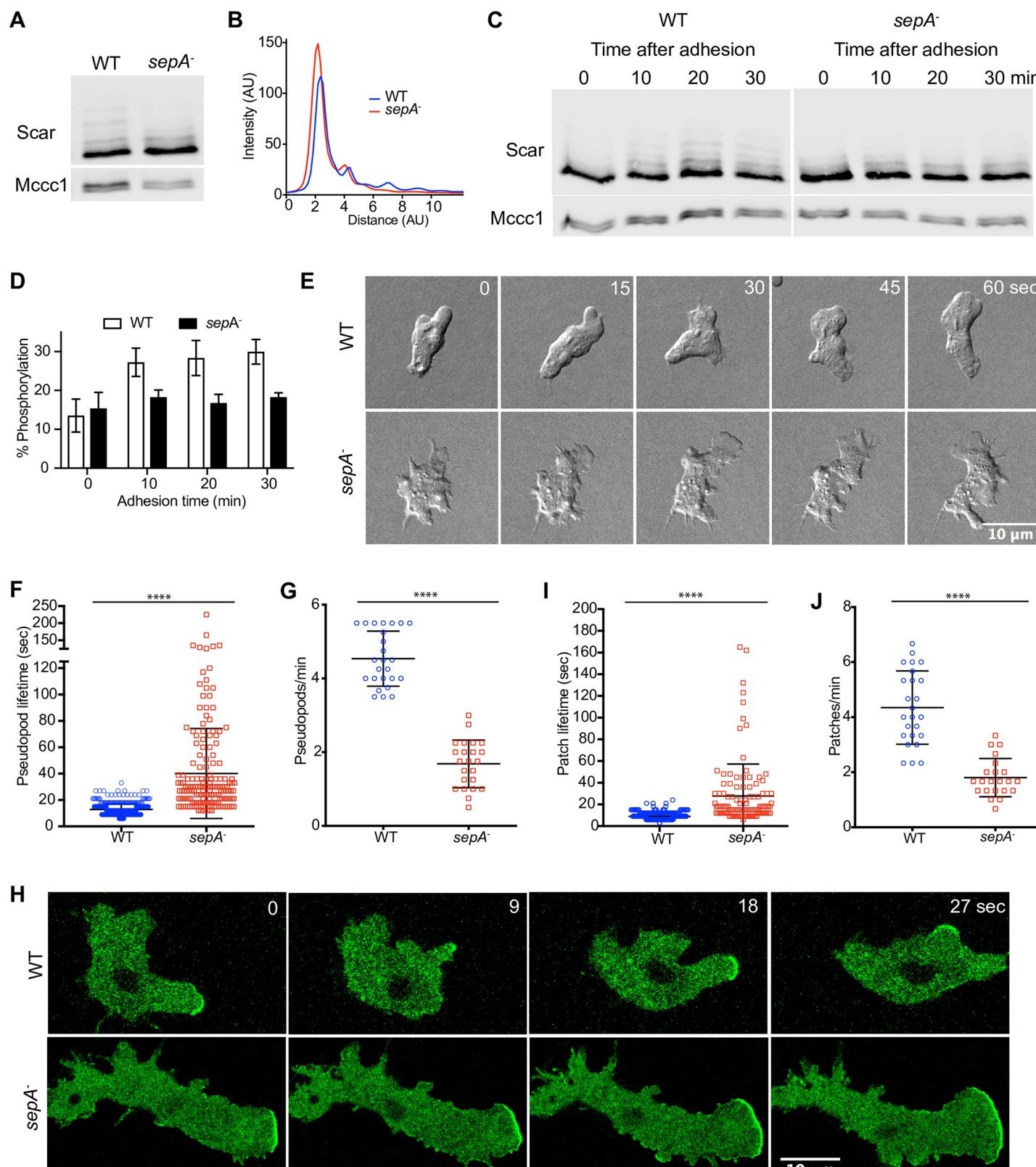

**Fig 8. SepA is required for Scar phosphorylation. (A)** Scar phosphorylation in WT and *sep*A⁻ cells. Cells were lysed, separated on low-bis gels, and probed for Scar. Higher bands are selectively lost. **(B)** Intensity plot from **(A)** showing reduction in number and intensity of higher peaks for *sep*A⁻ cells. **(C)** Acute effect of adhesion on

Scar/WAVE phosphorylation in WT and *sep*A⁻ cells. Cells growing in suspension were allowed to adhere in petri dishes and lysed at indicated time points. Cell lysates were analyzed by western blotting using low (0.06%) bis-acrylamide gels. The number and intensity of bands increases with time of adhesion only in WT and remains unchanged in *sep*A⁻ cells. **(D)** Quantitation of lane intensities (mean ± SD, $n = 3$) shows an increase in the intensities of phosphorylated Scar bands in WT and not in *sep*A⁻ cells. The numerical data are included in S1 Data. **(E,F,G)** Loss of *sep*A⁻ phenocopies expression of Scar$^{S8A}$. Cells were allowed to migrate under agarose up a folate gradient while being observed by DIC microscopy at a frame interval of 3 seconds (1f/3s). **(E)** *sep*A⁻ cells formed broader and more elongated pseudopods than WT. **(F)** Pseudopod lifetime, measured from DIC time-lapse videos. **(G)** Frequency of pseudopod formation. (mean ± SD; $n = 459^{WT}$, $168^{sepA-}$ over 3 independent experiments; $^{****}p \leq 0.0001$, $^{*}p \leq 0.05$, 2-tailed unpaired *t* test). The numerical data are included in S1 Data. **(H,I,J)**. Scar complex localization in *sep*A⁻ cells. **(H)** WT and *sep*A⁻ cells cells expressing HSPC300GFP were allowed to migrate under agarose up a folate gradient and examined by AiryScan confocal microscopy. Scar complex patches are broader **(H)** and longer-lived **(I)** in *sep*A⁻ cells but generated less frequently **(J)**. The numerical data are included in S1 Data. AU, arbitrary unit; DIC, differential interference contrast; F-actin, filamentous actin; WASP, Wiskott-Aldrich Syndrome protein; WT, wild type.

The kinase sites around the polyproline domain have little in common with one another and do not match the consensus sequences for most kinases that are active in the actin domain. We hypothesize that the phosphorylation sites are sterically inaccessible until the Scar/WAVE complex is activated. Once the complex is active it adopts an open conformation, whereupon SepA in particular (and maybe other constitutive kinases) can act on it. This means the phosphorylation is a passive reflection of the fact that the complex has been activated, rather than a driver or even modulator of the initial activation process. Of course, different kinases might have different effects in vivo, so our experiments could miss subtle controls, though the general conclusion that phosphorylation is inhibitory seems sound. The Ste20 family of kinases, part of the large and diverse MAPKKK (or MAP3K) family, is often associated with signaling pathways. The 2 most related family members in human, MAP3K19 and MAP3K3, are not well studied, so little can be concluded about their regulation; a number of other MAP3K family members are also very similar, so redundancy is possible. When phosphorylating *Dictyostelium* Scar/WAVE, SepA does not seem to require upstream activation, at least by G-protein linked or chemoattractant signaling, which paints a different picture of Scar/WAVE activation from what we had expected.

This work showcases our ability to use *scar⁻*/*wasp⁻* cells to test the functioning of mutated Scar proteins and the phenotypic changes caused by alterations in the sequence. Previously, results have been complicated because WASP changes its organization and cell biological role when the Scar complex is not present [8]. We have therefore been working against an ill-defined background of partial rescue. In this work, we show that either phosphomimetic or phosphorylation-deficient Scar is fully able to rescue the mutant phenotypes, because *scar⁻*/*wasp⁻* *Dictyostelium* cells cannot grow, do not make pseudopods, and barely move. Similarly, the use of WAVE1/WAVE2 knockout B16-F1 cells allows detailed study of physiologically normal levels of WAVEs without partial complementation. We therefore look forward to a new focus on physiological function and a more direct connection between the dynamics of the Scar/WAVE complex, the behavior of pseudopods and lamellipods, and cell migration.

Our results imply that cell:substrate adhesion is a crucial driver of Scar/WAVE activation. Adhesion has been an undeservedly neglected area of Scar/WAVE biology. It seems intuitively obvious that pseudopod formation should be regulated by adhesion—if pseudopods or lamellipods are generated in a location where adhesion is difficult or impossible, they will be inefficient or ineffective at producing cell migration. Cells should, in retrospect, therefore be expected to monitor adhesion and to favor pseudopods that can attach. The results in Fig 4 showing normal Scar/WAVE activation in talin knockouts offer 1 reason why connections have so far been missed—activation does not require the integrin-linked signaling pathways that dominate adhesion of (for example) mammalian fibroblasts [55]. Instead, we hypothesize that the connection is physical, by direct mechanical coupling. This is a well-documented and common mechanism for coupling proteins and cell behavior [56]. Overall, this work exposes a

new and conserved mechanism for regulating migration, which will influence cells moving in many physiological contexts. We look forward to seeing new biological examples emerge.

## Materials and methods

### Reagents and cell lines

All the reagents and cell lines used in this study are described in S2 Table. *Dictyostelium discoideum* strain AX3, haploid, mating type *mat*A derived from wild-type NC4 was obtained from *Dictyostelium* stock center [57].

### Growth of mammalian cells

Mammalian cells were grown in DMEM supplemented with 10% fetal bovine serum (FBS) or donor bovine serum (DBS) and 1% L-glutamine in tissue culture dishes at 37°C, 5% $CO_2$. Cells were washed twice with PBS prior to trypsinisation or lysis.

### Generation of WAVE1/2 knockout cells by CRISPR/Cas9-mediated genome editing

B16-F1 cells lacking *WASF1* and *WASF2* genes, encoding WAVE1 and WAVE2 proteins, respectively, were generated using CRISPR/Cas9 [5,58]. Specifically, WAVE2 KO cells were generated first by targeting exon 1 of the *WASF2* gene (GTGCCTTGGCTCGATGTTCC/ TGG) using pSpCas9(BB)-2A-Puro vector (Addgene plasmid ID: 48139). Subsequently, WAVE1/2 double KO cells were obtained upon transfection of WAVE2 KO clone #11 with pSpCas9(BB)-2A-Puro vector targeting the first coding exon of *WASF1* (GGCTGAGCTCAA GATGCCGT/TGG). Generation of knockout cell clones was initiated by addition of puromycin selection medium for 3 days after transfection with gene disruption vector, to remove nontransfected cells, followed by extensive dilution into 10-cm dishes and picking of macroscopically visible colonies several days later, to obtain single cell-derived clones. Clones were screened for the absence of respective gene product using WAVE1- and WAVE2-specific antibodies. Successful gene disruption was further validated by confirmation of the absence of any wild-type allele by sequencing of respective genomic regions.

### Inhibition of Rac1 activity by EHT1864

To determine the effects of Rac1 inhibition on Scar phosphorylation in *Dictyostelium*, EHT1864 (50 μM) was added to the medium or DB. Cells were lysed at 0, 2, 5, and 10 minutes after addition of EHT-1864 using LDS sample buffer (Thermo Fisher scientific) and analyzed for Scar phosphorylation. A total of $10^5$ cells (in Lo-Flo medium) expressing Pak-CRIB-mRFPmars 2 were seeded on Lab-Tek II chambered coverglasses (Thermo Fisher scientific). Cells were imaged by AiryScan confocal microscopy during and after treatment.

Similarly, to determine the effect of Rac1 inhibition on WAVE2 phosphorylation, B16-F1 cells were incubated with 50 μM EHT-1864 in DMEM (with 10% FBS and 1% L-glutamine) and lysed with RIPA buffer after 0, 2, 4, and 6 hours. The effect of Rac1 inhibition on lamellipods was determined by phalloidin immunofluorescence staining.

### Inhibition of actin polymerization by latrunculinA

Latrunculin A (5 μM) was added to the medium or DB. Cells were lysed at 0, 10, 20, and 30 minutes after addition of latrunculin and analyzed for Scar phosphorylation using low-bis acrylamide SDS-PAGE. To examine the effect of latrunculin on Scar/WAVE complex activation, $1 \times 10^5$ EGFP-NAP1 cells (in Lo-Flo medium) were seeded on Lab-Tek II chambered

coverglasses (Thermo Fisher Scientific). Cells were imaged by AiryScan confocal microscopy during and after treatment.

## Inhibition of Erk2 activation by U0126

To determine the effect of Erk2 inhibition on WAVE2 phosphorylation, growing B16-F1 and NIH3T3 cells were incubated with U0126 (10 μM) and cells were lysed with RIPA buffer at 0, 5, 10, 20, 30, and 60 minutes. Protein samples were analyzed for WAVE2 phosphorylation by low-bis acrylamide SDS-PAGE and western blotting.

## Transfection of *Dictyostelium* cells

Extrachromosomal plasmids were introduced into the cells by electroporation. A total of $1 \times 10^7$ cells/ml were washed one time with electroporation buffer (EB; 5 mM $Na_2HPO_4$, 5 mM $KH_2PO_4$, and 50 mM sucrose), and 0.5 μg plasmid DNA was electroporated into the cells by pulsing once at 500 V using an ECM399 electroporator (BTX Harvard). Cells were then transferred into HL5 medium. After 24 hours, 10 μg/ml G418 or 50 μg/ml hygromycin were added to select transformants.

## Transfection of mammalian cells

Extrachromosomal expression plasmids were transfected into B16-F1, WAVE1/2 knockout cells using lipofectamine-2000 according to manufacturer's instruction. In 6-well plates, $3 \times 10^5$ cells were allowed to adhere for 4–6 hours. Next, 7 μl lipofectamine-2000 and 2.5 μg plasmid DNA diluted and mixed in 200 μl serum free DMEM in 2 separate tubes. Later, diluted plasmid DNA and lipofectamine were pooled together and mixed well and incubated at room temperature (10 minutes). This mixture was added to the cells and mixed by swirling the plate. Then, 40 hours after transfection, cells were used for western blotting and immunofluorescence staining. Time-lapse and AiryScan imaging was performed after 24 hours of transfection.

## Stimulation of cells with folate/cAMP

A total of $2 \times 10^7$ cells/ml were incubated in DB for 1 hour, then 50 μM folate was added. Cells were lysed in LDS sample buffer at 0, 20, 40, 60, 120, and 240 seconds and tested by western blotting.

For cAMP stimulation, $2 \times 10^7$ cells/ml cells in DB were starved. After 1 hour, cells were pulsed with 100 nM cAMP every 6 minutes for 4 hours. After 15 minutes of last pulse cells were stimulated with 1 μM of cAMP and lysed in 1X LDS sample buffer at 0, 20, 40, 60, 120, and 240 seconds and used for western blotting.

## Adhesion and de-adhesion of *Dictyostelium* cells

*Dictyostelium* cells were grown either in suspension or in petri plates. Cells grown in adhesion were de-adhered by trituration with a P1000. For facilitating adhesion, $2 \times 10^6$ cells grown in suspension were added to 35-mm petri plates, lysed with LDS at appropriate time points, and western blotted.

## Immunofluorescence

B16-F1 and WAVE1/2 knockout cells transiently expressing WAVE2 constructs were allowed to adhere and form lamellipods on laminin A–coated coverslips for 4 hours, then washed twice with PBS and fixed in PBS/4% paraformaldehyde for 15 minutes. Blocking and

permeabilization were performed for 15 minutes in blocking solution (2% BSA, 0.1% Triton X-100 in PBS). Cells were stained with TRITC-phalloidin (1:200 in blocking solution) for 1 hour, washed twice in PBS, and mounted on slides using DAPI Fluoromount-G (Southern Biotech). After 24 hours, slides were imaged by AiryScan confocal.

### Under-agarose chemotaxis

Formation of pseudopods and cell migration were measured by under-agarose folate chemotaxis assay as described in Laevsky and Knecht, 2003 [59]. In brief, 0.4% SeamKem GTG agarose in Lo-Flo medium (ForMedium) was prepared by boiling. After cooling, 10 μM folic acid was added. Next, 5 ml of agarose-folate mix was poured into the BSA-coated 50-mm glass-bottom dishes (MatTek). A 5-mm-wide trough was cut with sterile scalpel and filled with $2\times10^6$ cells/ml. Cell migration was imaged after 4–6 hours with 10× and 60× DIC. To examine the localization of labeled proteins in the pseudopods, cells were also imaged by AiryScan confocal microscope.

### Western blotting

*Dictyostelium* cells were lysed by directly adding NuPAGE LDS sample buffer (Invitrogen) containing 20 mM DTT, HALT protease, and phosphatase inhibitors (Thermo Fisher Scientific) on top of moving cells and immediately boiling for 5 minutes. Mammalian cells were lysed with RIPA buffer, stored on ice for 10 minutes and debris cleared by centrifugation at 13,000 rpm, 5 minutes. Proteins were separated on 10% Bis-Tris NuPAGE gels (Invitrogen) or on hand-poured low-bis acrylamide (0.06% bis-acrylamide and 10% acrylamide) gels then separated at 150 V, 90 minutes. Proteins were electrotransferred onto 0.45 μM nitrocellulose membrane. Membranes were blocked in TBS+5% nonfat milk. Primary antibodies were used at 1:1,000 dilution. Fluorescently conjugated secondary antibody was used at 1:10,000 dilution to detect the protein bands by Odyssey CLx Imaging system (LI-COR Biosciences). For *Dictyostelium* samples Mccc1 [60] and for mammalian samples tubulin were used as a loading controls.

### Quantification of western blots

To quantify the proportion of Scar/WAVE phosphorylation, the total intensity of all bands (a) and the lowest band (b) were determined using the Odyssey CLx Imaging system (LI-COR Biosciences). The percentage phosphorylation is calculated by following formula:

$$\%\text{phosphorylation} = (a - b) \times 100/a.$$

### Phosphatase treatment

*Dictyostelium* cells grown in a 35-mm petri dish were lysed in 100 μL TN/T buffer (10 mM Tris-HCl [pH 7.5], 150 mM NaCl, and 0.1% Triton X-100) and kept on ice for 5 minutes. Mammalan cells were lysed in above buffer containing 1% triton X-100 and kept on ice for 5 minutes. Lysates were cleared by centrifugation (13,000 rpm, 4°C, 5 minutes) and proteins were phosphatased using 1 μL Lambda phosphatase at 30°C (NEB; P0753S). Western blotting assessed dephosphorylation of proteins as described.

### Cross-linking, GFP-TRAP pull-down, and mass spectrometry

Cell lysis and cross-linking was performed as described in [4].

Cells grown adhering to a 15-cm dish were washed with ice cold DB and lysed with 1 ml TNE/T buffer (10 mM Tris-HCl [pH 7.5], 150 mM NaCl, 0.5 mM EDTA and 0.1% Triton X-

100) containing HALT protease and phosphatase inhibitors. Lysates were kept at ice for 5 minutes. Lysates were cleared by centrifugation (13,000 rpm, 4˚C, 5 minutes). A volume of 25 μL GFP-TRAP beads (ChromoTek) were washed twice with TNE/T buffer and resuspended in 100 μL TNE/T buffer. Further, 900 μL of lysate was added to the beads and kept on rotation for 2 hours at 4˚C. Beads were spun down at (2,700$g$, 4˚C, 2 minutes) and washed 4 times with TNE/T buffer. To elute the proteins from the beads, 25 μL 2 x NuPAGE LDS sample buffer was added and boiled (100˚C, 5 minutes). Protein samples were analyzed on low-bis 10% acrylamide gels and proteins bands near Scar were analyzed by LC-MS/MS to identify phosphorylation sites on Scar.

## DNA constructs

To construct Scar/WAVE expression vector, Scar was PCR amplified from the *Dictyostelium discoideum* genomic DNA and cloned in pDM304 expression vector [61]. Scar/LifeAct-mRFPmars2 co-expression constructs were generated by ligating pDM641 NgoMIV fragment into Scar expression vectors. Scar phosphorylation sites were mutated by site-directed mutagenesis using primers mentioned in S3 Table. All mutations were confirmed by sequencing.

WAVE2 was amplified from pEGFP-WAVE2 and cloned into the pCDNA3.1 or pEGFP-C1 vector using KpnI and XbaI restriction sites. Phosphorylation sites were mutated to unphosphorylatable alanine by site-directed mutagenesis using primers listed in S3 Table.

## Mass spectrometry

**In-gel proteolytic digestion.** Eluates from GFP-NAP1 immunoprecipitation were separated by low-bis acrylamide SDS-PAGE and stained with Coomassie brilliant blue. The area of migration of SCAR indicated in S1A Fig was excised and digested separately with trypsin or chymotrypsin as previously described [62].

**MS analysis.** Tryptic peptides were separated by nanoscale C18 reverse-phase liquid chromatography using an EASY-nLC 1200 (Thermo Fisher Scientific) coupled online to an Orbitrap Q-Exactive HF mass spectrometer (Thermo Fisher Scientific) via nanoelectrospray ion source (Thermo Fisher scientific). Peptides were separated over a 60-minute gradient on a 20-cm fused silica emitter (New Objective) packed in house with reverse-phase Reprosil Pur Basic 1.9 μm (Dr. Maisch GmbH). For the full scan, a resolution of 60,000 at 250 Th was used. The top 10 most intense ions in the full MS were isolated for fragmentation with a target of 50,000 ions at a resolution of 15,000 at 250 Th. MS data were acquired using the XCalibur software (Thermo Fisher Scientific).

**Data analysis.** Raw data obtained were processed with MaxQuant version 1.5.5.1 [63]. Andromeda peak list files (.apl) generated were converted to Mascot generic files (.mgf) using APL to MGF Converter (The Walter and Eliza Hall institute of medical research, Australia; http://www.wehi.edu.au/people/andrew-webb/1298/apl-mgf-converter). The resulting MGF files were searched using Mascot (Matrix Science, version 2.4.1), querying dictyBase [57] (12,764 entries) plus an in-house database containing common proteomic contaminants and the sequence of GFP-NAP1 constructs used in the experiment. Mascot was searched assuming the digestion enzyme trypsin or chymotrypsin allowing for 2 miscleavages with a fragment ion mass tolerance of 50 ppm and a parent ion tolerance of 10 ppm. The iodoacetamide derivative of cysteine was specified in Mascot as a fixed modification, and oxidation of methionine and phosphorylation of serine, threonine, and tyrosine were specified in Mascot as variable modifications.

Scaffold (version 4.3.2, Proteome Software) was used to validate MS/MS-based peptide and protein identifications. Peptide identifications were accepted if they could be established at

greater than 95.0% probability as specified by the PeptideProphet algorithm, resulting in a peptide false discovery rate (FDR) of 0.63% [64]. The identified phosphopeptides are listed in S4 Table.

**Microscopy and image analysis.** To determine the morphology, chemotactic speed, and directionality of cells, phase contrast time-lapse microscopy was performed at 10x/0.3NA on Nikon ECLIPSE TE-2000-R inverted microscope equipped with Retiga EXI CCD monochromatic camera. Images of cells migrating under agarose toward folate gradient were captured per minute for 45 minutes. DIC images were taken every 3 seconds for 7.5 minutes with 60x/1.4 NA to observe the pseudopod formation. The lifetime and width of pseudopods were quantified by ImageJ. EGFP-Nap1- or HSPC300-EGFP-expressing *Dictyostelium* cells were used to determine the activation of Scar complex. The localization of Scar/WAVE complex, Pak-CRIB, and F- Actin used a 63x/1.4 NA objective on an AiryScan Zeiss 880 inverted confocal microscope.

Images were analyzed by homemade ImageJ plugin written by Dr. Luke Tweedy to determine chemotactic speed and tortuosity.

Random motility assay was performed to determine the velocity, lamellipods' lifetime, and width of WAVE1/2 knockout cells expressing pCDNA3.1, WAVE2$^{WT}$, and WAVE2$^{S8/T1A}$. Cells were seeded on laminin-coated glass-bottom 6-well plates and allowed to adhere and form lamellipods for 4 hours. Time-lapse movies were recorded at the interval of 2.5 or 10 minutes for 16 hours. Movies were analyzed using the MTrackJ plugin to calculate the velocity of cells. The resulting data was analyzed by chemotaxis tool. Lamellipod width and perimeter of cells were measured by straight or freehand line measurements.

EGFP-WAVE2$^{WT}$- and EGFP-WAVE2$^{S8/T1A}$-expressing WAVE1/2 knockout cells were seeded on laminin-coated 35-mm glass-bottom dishes and imaged by AiryScan super-resolution imaging as before.

## Quantification and statistical analysis

All the experiments were performed at least 3 times, and individual data were quantified to avoid any error. The cell's speed and tortuosity were calculated from hundreds of cells. Scar/WAVE patch dynamics was observed from hundreds of pseudopods of at least dozens of cells. Any data points were not excluded from the list at any time of analysis.

To compare the expression level and changes in phosphorylation of Scar, blots were quantified using Odyssey CLx Imaging system (LI-COR Biosciences) analysis tool.

Statistical significance analyses of manual measurements such as speed, tortuosity, Scar patch lifetime/patch accumulation size, and patch frequency were performed by nonparametric statistics, such as 1-way ANOVA and Dunn's multiple comparison tests. The experimental groups were tested for normal distribution using Shapiro–Wilk test of Prism 7 software (Graphpad, La Jolla, USA). Sample sizes are provided in the figure legends. *n* refers to biological repeats in western blotting, total number of cells for cell speed, tortuosity and Scar patch frequency, duration of patches/accumulation size in $\geq 25$ or more cells.

## Supporting information

**S1 Fig. GFP-TRAP pull-down of EGFP-NAP1 cells.** (A) Coomassie brilliant blue stained low-bis acrylamide PAGE gel of GFP-TRAP pull-down samples. Lane 1, 2, and 3 indicates samples from Scar-/EGFPNAP1, and lane 4, 5, and 6 indicates samples from EGFPNap1 cells. The Scar band indicated on the gel was excised for LC-MS/MS. (B) Representative western blot of above gel indicating phosphorylated Scar bands. LC-MS/MS, liquid chromatography-

tandem mass spectrometry.
(PDF)

**S2 Fig. Incorporation of Scar mutants in the complex.** (A) GFP-TRAP pull-down was performed using MG132-treated and untreated *Dictyostelium* Scar-/EGFP-NAP1 cells expressing Scar$^{WT}$, Scar$^{S8A}$, and Scar$^{S8D}$. The eluates were analyzed on low-bis PAGE western blotting for Pir121/Nap1/Scar and ubiquitin. (B) Scar expression in the cell lysates was analyzed by low-bis PAGE and western blotting.
(PDF)

**S3 Fig. Phosphorylation of Scar during starvation.** Cells were washed with non-nutrient buffer, starved for the indicated times, separated on low-bis gels, and probed with anti-Scar. The amount of phosphorylated Scar is reduced following development.
(PDF)

**S4 Fig. Expression pattern and effect of endogenous Scar, Scar$^{WT}$, Scar$^{S8A}$, and Scar$^{8D}$ in the complex formation.** Nap-/EGFP-Nap1 and Scar-/Nap-/EGFP-Nap1 cells rescued with Scar$^{WT}$, Scar$^{S8A}$, and Scar$^{8D}$ cell lysates were immunoprecipitated using GFP-TRAP. (A–C) Lysate and pull-down samples were analyzed for the expression of Pir121, Nap1, Scar, and Abi. Quantification of western blots shows that similar to Scar$^{Endo}$, Scar$^{WT}$, Scar$^{S8A}$, and Scar$^{8D}$ formed stable complexes. The numerical data are included in S1_Data. (D–E) Phosphorylated Scar in the complex. Lysate and GFP-TRAP samples were analyzed on low-bis gels. Scar$^{Endo}$ and Scar$^{WT}$ are similarly phosphorylated in lysates and GFP-TRAP samples. The numerical data are included in S1 Data.
(PDF)

**S1 Video. Effect of EHT1864 on Rac1 and Scar complex localization.** Dictyostelium cells expressing PakCRIB-mRFPmars2 were allowed to migrate under agarose up folate gradient and observed by AiryScan confocal microscopy. Filmed at 1 frame/2 seconds, movie shows 10 frames/second. EHT1864 was added at frame 7 (after 14 seconds) in the video.
(MOV)

**S2 Video. Scar/WAVE and Rac1 activation in cells with mutant PIR121 A site.** Pir121 knockout cells expressing WT Pir121-EGFP and A-site Pir121-EGFP were further expressed with PakCRIB-mRFPmars2. Scar complex (green) and PakCRIB-mRFPmars2 (red) localization was visualized in migrating cells under agarose up folate gradient. Filmed at 1 frame/2 seconds, Movie shows 10 frames/second.
(MOV)

**S3 Video. Effect of Latrunculin treatment on Scar complex localization.** eGFP-NAP1 cells were seeded on Lab-Tek II coverglass chambers and imaged by AiryScan imaging. LatrunculinA (5 μm) was added to the cells undergoing imaging. Filmed at 1 frame/30 seconds, movie shows 10 frames/second. Latrunculin was added after frame 2 (after 1 minute).
(MOV)

**S4 Video. Pseudopod formation in Scar phosphomutants.** Scar- cells expressing Scar$^{WT}$, Scar$^{S8A}$, and Scar$^{S8D}$ were allowed to migrate under agarose up a folate gradient and observed by DIC. Filmed at 1 frame/3 seconds, movie shows 10 frames/second.
(MOV)

**S5 Video. Scar complex localization in Scar phosphomutants.** Scar-/EGFP-Nap1 cells expressing Scar$^{WT}$, Scar$^{S8A}$, and Scar$^{S8D}$ were allowed to migrate under agarose up folate gradient, and Scar complex activation in pseudopods were observed by AiryScan confocal

microscopy. Filmed at 1 frame/3 seconds, movie shows 10 frames/second.
(MOV)

**S6 Video. Scar complex activation in total Scar phosphomutants.** Scar-/EGFP-Nap1 cells expressing Scar$^{S13A}$ and Scar$^{S13D}$ were allowed to migrate under agarose up folate gradient and Scar complex activation in pseudopods were observed by AiryScan confocal microscopy. Filmed at 1 frame/3 seconds, movie shows 10 frames/second.
(MOV)

**S7 Video. Recruitment of WAVE complex and lamellipod formation in WAVE1/2 KO rescued with WAVE2$^{WT}$ and WAVE2$^{S8A/T1A}$.** Randomly migrating cells were imaged using AiryScan confocal microscopy. Filmed at 1 frame/20 seconds, movie shows 5 frames/second.
(MOV)

**S8 Video. Scar complex activation in Scar-/wasp- cells expressing phosphomutant Scar.** Scar$^{tet}$/wasp- cells expressing Scar$^{WT}$, Scar$^{S8A}$, and Scar$^{S8D}$ were allowed to migrate under agarose up folate gradient and Scar complex activation in pseudopods were observed by AiryScan confocal microscopy. Filmed at 1 frame/2 seconds, movie shows 10 frames/second.
(MOV)

**S9 Video. Pseudopod formation in Scar-/wasp- cells expressing phosphomutant Scar.** Scar$^{tet}$/wasp- cells expressing Scar$^{WT}$, Scar$^{S8A}$, and Scar$^{S8D}$ were allowed to migrate under agarose up folate gradient and were observed by differential interference contrast microscopy. Filmed at 1 frame/2 seconds, movie shows 10 frames/second.
(MOV)

**S10 Video. Pseudopod formation in WT and sepA- cells.** WT and sepA- were allowed to migrate under agarose up folate gradient and were observed by differential interference contrast microscopy. Filmed at 1 frame/3 seconds, movie shows 10 frames/second. WT, wild type.
(MOV)

**S11 Video. Subcellular localization Scar complex in WT and sepA-.** WT and sepA- expressing HSPC300-GFP. were allowed to migrate under agarose up folate gradient and were observed by AiryScan confocal micoscopy. Filmed at 1 frame/3 seconds, movie shows 10 frames/second. WT, wild type.
(MOV)

**S1 Raw images. Unprocessed images of all gels and blots in the paper.**
(PDF)

**S1 Data. Numerical data for Fig 2E and 2H; Fig 3B, 3C, 3E, 3F, 3H, 3L and 3N; Fig 4D and 4F; Fig 5B, 5C, 5D, 5E, 5F and 5G; Fig 6B, 6C, 6D, 6G, 6H, 6I and 6J; Fig 7B and 7F; Fig 8D, 8F, 8G, 8I and 8J and S4B, S4C and S4E Fig.** (xlsx).
(XLSX)

**S1 Table Peptide sequences and highlighted phosphorylated residues.**
(DOCX)

**S2 Table. List of reagents, cells, and plasmids used in this study.**
(DOCX)

**S3 Table. List of oligos used in Scar/WAVE cloning and mutagenesis of unphospho- Scar and phosphomimetic Scar.**
(DOCX)

**S4 Table. List of phosphopeptides detected by LC-MS/MS.** LC-MS/MS, Liquid chromatography-tandem mass spectrometry.
(DOCX)

## Acknowledgments

We greatly acknowledge dictyBase (www.dictybase.org) and the Dictyostelium stock center for sending strains. We thank Prof. Robert R. Kay for *erk*A and *erk*B knockout cells; Dr. Rachana Patel for providing NIH3T3 cells; Drs Margaret O'Prey, Heather Spence, and Jamie Whitelaw for help in microscopy and mammalian cell culture; Luke Tweedy for quantification of cell parameters; and Dr. Simona Buracco for discussions. We also thank Prof. Annette Muller-Taubenberger for discussions about *sep*A mutant phenotypes.

## Author Contributions

**Conceptualization:** Shashi Prakash Singh, Peter A. Thomason, Laura M. Machesky, Robert H. Insall.

**Data curation:** Shashi Prakash Singh.

**Formal analysis:** Robert H. Insall.

**Funding acquisition:** Robert H. Insall.

**Investigation:** Shashi Prakash Singh.

**Methodology:** Shashi Prakash Singh, Sergio Lilla, Matthias Schaks, Qing Tang, Klemens Rottner, Robert H. Insall.

**Project administration:** Robert H. Insall.

**Resources:** Matthias Schaks, Bruce L. Goode, Klemens Rottner, Robert H. Insall.

**Supervision:** Robert H. Insall.

**Validation:** Shashi Prakash Singh.

**Writing – original draft:** Shashi Prakash Singh, Robert H. Insall.

**Writing – review & editing:** Bruce L. Goode, Laura M. Machesky, Robert H. Insall.

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
