## [Decision Letter · Decision Letter 0]

14 May 2020

Dear Robert, 

I hope everything is going well and please accept my apologies for the delay in this manuscript. We are completely slammed at the moment and have half of the team working part time to be able to home school kids, so we are trying to cope as best as we can.

We are now consulting with the academic editor on the decision, but I've just realised that this was a new submission and that I completely forgot to ask you to complete again all the metadata before sending the manuscript for review. Would you mind to do this now? Otherwise the system won't allow me to send the decision.

Please login to Editorial Manager where you will find the paper in the 'Submissions Needing Revisions' folder on your homepage. Please click 'Revise Submission' from the Action Links and complete all additional questions in the submission questionnaire. And just re-submit the manuscript as soon as you can.

Once your full submission is complete, your paper will undergo a series of checks and once they are all fine, the manuscript will be ready for the decision.

Please let me know if you encounter any problems and apologies for the oversight!

Best wishes,

Ines

--

Ines Alvarez-Garcia, PhD

Senior Editor

PLOS Biology

Carlyle House, Carlyle Road

Cambridge, CB4 3DN

+44 1223–442810

---

## [Editor Report · Decision Letter 1]

18 May 2020

Dear Robert,

Thank you for submitting your revised Initial Research Submission entitled "Cell-substrate adhesion drives Scar/WAVE activation and phosphorylation, which controls pseudopod lifetime" for publication in PLOS Biology. Thank you also for your patience as we completed our editorial process, and please accept again my apologies for the delay in providing you with our decision. I have now obtained advice from three of the original reviewers and have discussed their comments with the Academic Editor. 

Based on the reviews (attached below), we will probably accept this manuscript for publication, assuming that you will modify the manuscript to address the remaining points raised by Reviewer 3. Please also make sure to address the data and other policy-related requests noted at the end of this email.

We expect to receive your revised manuscript within two weeks. Your revisions should address the specific points made by each reviewer. In addition to the remaining revisions and before we will be able to formally accept your manuscript and consider it "in press", we also need to ensure that your article conforms to our guidelines. A member of our team will be in touch shortly with a set of requests. As we can't proceed until these requirements are met, your swift response will help prevent delays to publication.

*Copyediting*

*Published Peer Review History*

*Early Version*

*Submitting Your Revision*

Best wishes,

Ines

--

Ines Alvarez-Garcia, PhD

Senior Editor

PLOS Biology

Carlyle House, Carlyle Road

Cambridge, CB4 3DN

+44 1223–442810

DATA POLICY:

Fig. 2E, H; Fig. 3B, C, E, F, H, L, N; Fig. 4D, F; Fig. 5B, C, D, E, F, G; Fig. 6C, D, G, H, I, J; Fig. 7B, F; Fig. 8D, F, G, I, J and Fig. S4B, C, E

Reviewers’ comments

Rev. 1:

The authors have appropriately responded and addressed the reviewers' comments and concerns.

Rev. 3:

My major concern about the first manuscript version was the quite limited novelty of the findings, particularly the missing functionally relevant serine kinase that phosphorylates SCAR/WAVE in the described cell adhesion-dependent context. In the revised version the Insall group now provided strong evidence that the Ste20 group kinase, SepA contributes substantially to SCAR/WAVE phosphorylation, an exciting novel finding that strengthens the whole story.

Thus overall, I am satisfied with the improved version of the manuscript. I only have two remaining minor points. First, the authors should include the SepA kinase into manuscript title. Secondly, the authors should further discuss their findings and possible conserved functions of Ste20 like kinases in SCAR/WAVE regulation in more detail.

Rev. 4:

In the revised version of this manuscript, the authors have somewhat improved the presentation of their story by eliminating some of the more poorly conceived experiments and adding some new data such as the identification of an orphan kinase that phosphorylates Scar in Dicty cells. In the end however, this story is just not that interesting. They accomplish two things: 1. showing the Erk is not a major kinase for Scar/Wave and 2. Phosphorylation has a minor effect on pseudopod stability that does not apparently have much affect on whole cell motility or the fidelity of directed migration. While aficionados of Scar/Wave biology may be interested in this, I doubt that this will really have much influence on the field of actin biology or motility.

---

## [Editor Report · Decision Letter 2]

13 Jul 2020

Dear Dr Insall,

On behalf of my colleagues and the Academic Editor, Cornelis Weijer, I am pleased to inform you that we will be delighted to publish your Research Article in PLOS Biology. 

Early Version

PRESS 

Kind regards,

Alice Musson

Publishing Editor, 

PLOS Biology

on behalf of

Ines Alvarez-Garcia,

Senior Editor

PLOS Biology